# Replicable Clustering[*]

**Hossein Esfandiari**
Google Research
esfandiari@google.com

**Amin Karbasi**
Yale University, Google Research
amin.karbasi@yale.edu

**Vahab Mirrokni**
Google Research
mirrokni@google.com

**Grigoris Velegkas**
Yale University
grigoris.velegkas@yale.edu

**Felix Zhou**
Yale University
felix.zhou@yale.edu

## Abstract

We design replicable algorithms in the context of statistical clustering under the recently introduced notion of replicability from Impagliazzo et al. [2022]. According to this definition, a clustering algorithm is replicable if, with high probability, its output induces the *exact* same partition of the sample space after two executions on different inputs drawn from the same distribution, when its *internal* randomness is shared across the executions. We propose such algorithms for the statistical $k$-medians, statistical $k$-means, and statistical $k$-centers problems by utilizing approximation routines for their combinatorial counterparts in a black-box manner. In particular, we demonstrate a replicable $O(1)$-approximation algorithm for statistical Euclidean $k$-medians ($k$-means) with $\tilde{O}(\text{poly}(k,d)k^{\log \log k})$ sample complexity. We also describe an $O(1)$-approximation algorithm with an additional $O(1)$-additive error for statistical Euclidean $k$-centers, albeit with $\tilde{O}(\text{poly}(k)\exp(d))$ sample complexity. In addition, we provide experiments on synthetic distributions in 2D using the $k$-means++ implementation from sklearn as a black-box that validate our theoretical results[2].

## 1 Introduction

The unprecedented increase in the amount of data that is available to researchers across many different scientific areas has led to the study and development of automated data analysis methods. One fundamental category of such methods is *unsupervised learning* which aims to identify some inherent structure in *unlabeled* data. Perhaps the most well-studied way to do that is by grouping together data that share similar characteristics. As a result, *clustering* algorithms have become one of the central objects of study in unsupervised learning. Despite a very long line of work studying such algorithms, e.g. Jain and Dubes [1988], Hart et al. [2000], Anderberg [2014], there is not an agreed-upon definition that quantifies the quality of a clustering solution. Kleinberg [2002] showed that there is an inherent reason why this is the case: it is impossible to design a clustering function that satisfies three natural properties, namely *scale-invariance*, *richness of solutions*, and *consistency*. This means that the algorithm designer needs to balance several conflicting desiderata. As a result, the radically different approaches that scientists use depending on their application domain can be sensitive to several factors such as their random initialization, the measure of similarity of the data, the presence of noise in the measurements, and the existence of outliers in the dataset. All these issues give rise to algorithms whose results are not *replicable*, i.e., when we execute them on two different samples of the same population, they output solutions that vary significantly. This begs the

---

[*]Authors are listed alphabetically.

[2]https://anonymous.4open.science/r/replicable_clustering_experiments-E380

37th Conference on Neural Information Processing Systems (NeurIPS 2023).

following question. Since the goal of clustering algorithms is to reveal properties of the underlying population, how can we trust and utilize their results when they fail to pass this simple test?

Replicability is imperative in making sure that scientific findings are both valid and reliable. Researchers have an obligation to provide coherent results and conclusions across multiple repetitions of the same experiment. Shockingly, a 2016 survey that appeared in Nature [Baker, 2016] revealed that 70% of researchers tried, but were unable to, replicate the findings of another researcher and more than 50% of them believe there is a significant crisis in replicability. Unsurprisingly, similar worries have been echoed in the subfields of machine learning and data science [Pineau et al., 2019, 2021]. In this work, we initiate the study of replicability in clustering, which is one of the canonical problems of unsupervised learning.

## 1.1 Related Works

**Statistical Clustering.** The most relevant previous results for (non-replicable) statistical clustering was established by Ben-David [2007], who designed $O(1)$-approximation algorithms for statistical $k$-medians ($k$-means) with $O(k)$ sample complexity. However, their algorithm picks centers from within the samples and is therefore non-replicable.

**Combinatorial Clustering.** The flavor of clustering most studied in the approximation algorithms literature is the setting where we have a uniform distribution over finite points and the algorithm has explicit access to the entire distribution. Our algorithms rely on having black-box access to a combinatorial clustering oracle. See Byrka et al. [2017], Ahmadian et al. [2019] for the current best polynomial-time approximation algorithms for combinatorial $k$-medians ($k$-means) in general metrics with approximation ratio 2.675 (9). Also, see Cohen-Addad et al. [2022] for a 2.406 (5.912) approximation algorithm for the combinatorial Euclidean $k$-medians ($k$-means).

**Clustering Stability.** Stability in clustering has been studied both from a practical and a theoretical point of view [Ben-Hur et al., 2001, Lange et al., 2004, Von Luxburg and Ben-David, 2005, Ben-David et al., 2006, Rakhlin and Caponnetto, 2006, Ben-David et al., 2007, Von Luxburg et al., 2010]. In most applications, it is up to the algorithm designer to decide upon the value of $k$, i.e., the number of different clusters. Thus, it was proposed that a necessary condition it should satisfy is that it leads to solutions that are not very far apart under *resampling* of the input data [Ben-Hur et al., 2001, Lange et al., 2004]. However, it was shown that this notion of stability for center-based clustering is heavily based on symmetries within the data which may be unrelated to clustering parameters [Ben-David et al., 2006]. Our results differ from this line of work in that we require the output across two separate samples to be *exactly the same* with high probability, when the randomness is shared. Moreover, our work reaffirms Ben-David et al. [2006] in that their notion of stability can be perfectly attained no matter the choice of $k$.

Other notions of stability related to our work include robust hierarchical clustering [Balcan et al., 2014], robust online clustering [Lattanzi et al., 2021], average sensitivity [Yoshida and Ito, 2022], and differentially private (DP) clustering [Cohen et al., 2021, Ghazi et al., 2020]. The definition of replicability we use is statistical and relies on an underlying data distribution while (DP) provides a worst-case combinatorial guarantee for two runs of the algorithm on neighboring datasets. Bun et al. [2023], Kalavasis et al. [2023] provide connections between DP and replicability for statistical learning problems. However, these transformations are not computationally efficient. It would be interesting to come up with computationally efficient reductions between replicable and DP clustering algorithms.

**Coresets for Clustering.** A long line of work has focused on developing strong coresets for various flavors of centroid-based clustering problems. See Sohler and Woodruff [2018] for an overview of this rich line of work. The most relevant for our results include coresets for dynamic geometric streams through hierarchical grids [Frahling and Sohler, 2005] and sampling based techniques [Ben-David, 2007, Feldman and Langberg, 2011, Bachem et al., 2018].

**Dimensionality Reduction.** Dimensionality reduction for clustering has been a popular area of study as it reduces both the time and space complexity of existing algorithms. The line of work on data-oblivious dimensionality reduction for $k$-means clustering was initiated by Boutsidis et al. [2010]. The goal is to approximately preserve the cost of all clustering solutions after passing the data through a dimensionality reduction map. This result was later improved and generalized to $(k, p)$-clustering [Cohen et al., 2015, Becchetti et al., 2019], culminating in the work of Makarychev

et al. [2019], whose bound on the target dimension is sharp up to a factor of $\log 1/\varepsilon$. While Charikar and Waingarten [2022] overcome this factor, their result only preserves the cost across the optimal solution.

**Replicability in ML.** Our results extend the recently initiated line of work on designing provably replicable learning algorithms under the definition that was introduced by Impagliazzo et al. [2022]. Later, Esfandiari et al. [2022] considered a natural adaption of this definition to the setting of bandits and designed replicable algorithms that have small regret. A slightly different notion of replicability in optimization was studied in Ahn et al. [2022], where it is required that an optimization algorithm that uses noisy operations during its execution, e.g., noisy gradient evaluations, outputs solutions that are close when executed twice. Subsequently, Bun et al. [2023], Kalavasis et al. [2023] established strong connections between replicability and other notions of algorithmic stability. Recently, Dixon et al. [2023], Chase et al. [2023] proposed a weaker notion of replicability where the algorithm is not required to output the same solution across two executions, but its output needs to fall into a small list of solutions.

## 2 Setting & Notation

Let $\mathcal{X} \subseteq \mathbb{R}^d$ be the instance space endowed with a metric $\kappa : \mathcal{X} \times \mathcal{X} \to \mathbb{R}_+$ and $\mathbb{P}$ be a distribution on $\mathcal{X}$ which generates the i.i.d. samples that the learner observes.

For $F \subseteq \mathbb{R}^d$ and $x \in \mathcal{X}$, we overload the notation and write $F(x) := \operatorname{argmin}_{f \in F} \kappa(x, f)$ to be the closest point to $x$ in $F$ as well as $\kappa(x, F) := \kappa(x, F(x))$ to be the shortest distance from $x$ to a point in $F$.

We assume that $\mathcal{X}$ is a subset of $\mathcal{B}_d$, the $d$-dimensional $\kappa$-ball of diameter 1 centered about the origin[3]. We also assume that $\kappa$ is induced by some norm $\|\cdot\|$ on $\mathbb{R}^d$ that is *sign-invariant* (invariant to changing the sign of a coordinate) and *normalized* (the canonical basis has unit length). Under these assumptions, the unit ball of $\kappa$ is a subset of $[-1, 1]^d$.

Our setting captures a large family of norms, including the $\ell_p$-norms, Top-$\ell$ norms (sum of $\ell$ largest coordinates in absolute value), and ordered norms (non-negative linear combinations of Top-$\ell$ norms) [Chakrabarty and Swamy, 2019]. Our results hold for more general classes of norms but for the sake of simplicity, we abide by these assumptions.

We define $\Delta := \sup\{\kappa(x, y) : x, y \in [0, 1)^d\}$ to be the $\kappa$-diameter of the unit hypercube. Note that $1 \le \Delta \le d$ by assumption. Moreover, $L\Delta$ is the $\kappa$-diameter of a hypercube with side length $L$. For example, if $\|\cdot\| = \|\cdot\|_2$ is the Euclidean norm, then $\Delta = \sqrt{d}$.

### 2.1 Clustering Methods and Generalizations

We now introduce the clustering objectives that we study in this work, which all fall in the category of minimizing a cost function $\operatorname{cost} : \mathcal{F} \to \mathbb{R}_+$, where $\mathcal{F} := \{F \subseteq \mathcal{B}_d : |F| = k\}$. We write $\operatorname{cost}(F)$ to denote the objective in the statistical setting and $\widehat{\operatorname{cost}}(F)$ for the combinatorial setting in order to distinguish the two.

**Problem 2.1** (Statistical $(k, p)$-Clustering)**.** Given i.i.d. samples from a distribution $\mathbb{P}$ on $\mathcal{X} \subseteq \mathcal{B}_d$, minimize $\operatorname{cost}(F) := \mathbb{E}_{x \sim \mathbb{P}} \kappa(x, F)^p$.

In other words, we need to partition the points into $k$ clusters so that the expected distance of a point to the center of its cluster, measured by $\kappa(\cdot, \cdot)^p$, is minimized. This is closely related to the well-studied combinatorial variant of the $(k, p)$-clustering problem.

**Problem 2.2** ($(k, p)$-Clustering)**.** Given some points $x_1, \ldots, x_n \in \mathcal{X}$, minimize $\widehat{\operatorname{cost}}(F) := \frac{1}{n} \sum_{i=1}^{n} \kappa(x_i, F)^p$.

We note that (statistical) $k$-medians and (statistical) $k$-means is a special case of Problem 2.2 (Problem 2.1) with $p = 1, 2$, respectively. We also consider a slight variant of the combinatorial problem, i.e., Problem 2.2, where we allow different points $x_i$ to participate with different weights $w_i$ in the objective. We refer to this problem as the *weighted* $(k, p)$-clustering problem.

---

[3]Our results can be generalized to $\kappa$-balls of diameter $L$ with an arbitrary center through translation and scaling.

We now shift our attention to the $k$-centers problem.

**Problem 2.3** (Statistical $k$-Centers). Given i.i.d. samples from a distribution $\mathbb{P}$ on $\mathcal{X} \subseteq \mathcal{B}_d$, minimize $\mathrm{cost}(F) := \max_{x \in \mathcal{X}} \kappa(x, F)$.

Notice that the $\ell_\infty$ norm is the limit of the $\ell_p$ norm as $p$ tends to infinity, hence $k$-centers is, in some sense, the limit of $(k, p)$-clustering as $p$ tends to infinity. Also, notice that this problem differs from $k$-means and $k$-medians in the sense that it has a *min-max* flavor, whereas the other two are concerned with minimizing some *expected* values. Due to this difference, we need to treat $k$-centers separately from the other two problems, and we need to make some assumptions in order to be able to solve it from samples (cf. Assumption F.1, Assumption F.2). We elaborate more on that later.

Let us also recall the combinatorial version of $k$-centers.

**Problem 2.4** ($k$-Centers). Given some points $x_1, \ldots, x_n \in \mathcal{X}$, minimize $\widehat{\mathrm{cost}}(F) := \max_{i \in [n]} \kappa(x_i, F)$.

We remark that clustering has mainly been studied from the combinatorial point of view, where the distribution is the uniform distribution over some finite points and we are provided the entire distribution. The statistical clustering setting generalizes to arbitrary distributions with only sample access. We emphasize that although we only have access to samples, our output should be a good solution for the entire distribution and not just the observed data.

We write $F_{\mathrm{OPT}}$ to denote an optimal solution for the entire distribution and $\mathrm{OPT} := \mathrm{cost}(F_{\mathrm{OPT}})$. Similarly, we write $\widehat{F}_{\mathrm{OPT}}$ to denote an optimal sample solution and $\widehat{\mathrm{OPT}} := \widehat{\mathrm{cost}}(\widehat{F}_{\mathrm{OPT}})$. Suppose we solve Problem 2.4 given a sample of size $n$ from Problem 2.3. Then $\widehat{\mathrm{OPT}} \leq \mathrm{OPT}$ since we are optimizing over a subset of the points.

Recall that a $\beta$-*approximate solution* $F$ is one which has cost $\mathrm{cost}(F) \leq \beta \, \mathrm{OPT}$. Note this is with respect to the statistical version of our problems. An algorithm that outputs $\beta$-approximate solutions is known as a $\beta$-*approximation algorithm*. We also say that $F$ is a $(\beta, B)$-*approximate solution* if $\mathrm{cost}(F) \leq \beta \, \mathrm{OPT} + B$.

### 2.1.1 Parameters $p$ and $\kappa$

Here we clarify the difference between $p$ and $\kappa$, which are two separate entities in the cost function $\mathbb{E}_x \left[ \kappa(x, F(x))^p \right]$. We denote by $\kappa$ the distance metric used to measure the similarity between points. The most commonly studied and applied option is the Euclidean distance for which our algorithms are the most sample-efficient. On the other hand, $p$ is the exponent to which we raise the distances when computing the cost of a clustering. A smaller choice of $p$ puts less emphasis on points that are far away from centers and $p = 1$ seeks to control the average distance to the nearest center. A large choice of $p$ puts emphasis on points that are further away from centers and as $p$ tends to infinity, the objective is biased towards solutions minimizing the maximum distance to the nearest center. Thus we can think of $k$-centers as $(k, p)$-clustering when $p = \infty$. As a concrete example, when $\kappa$ is the Euclidean distance and $p = 5$, the cost function becomes $\mathbb{E}_x \left[ \|x - F(x)\|_2^5 \right]$.

## 2.2 Replicability

Throughout this work, we study replicability[4] as an algorithmic property using the definition of Impagliazzo et al. [2022].

**Definition 2.5** (Replicable Algorithm; [Impagliazzo et al., 2022]). Let $\rho \in (0, 1)$. A randomized algorithm $\mathcal{A}$ is $\rho$-*replicable* if for two sequences of $n$ i.i.d. samples $\bar{X}, \bar{Y}$ generated from some distribution $\mathbb{P}^n$ and a random binary string $\bar{r} \sim R(\mathcal{X})$,

$$\mathbb{P}_{\bar{X}, \bar{Y} \sim \mathbb{P}^n, \bar{r} \sim R(\mathcal{X})} \{ \mathcal{A}(\bar{X}; \bar{r}) = \mathcal{A}(\bar{Y}; \bar{r}) \} \geq 1 - \rho, .$$

In the above definition, we treat $\mathcal{A}$ as a randomized mapping to solutions of the clustering problem. Thus, even when $\bar{X}$ is fixed, $\mathcal{A}(\bar{X})$ should be thought of as random variable, whereas $\mathcal{A}(\bar{X}; \bar{r})$ is the *realization* of this variable given the (fixed) $\bar{X}, \bar{r}$. We should think of $\bar{r}$ as the shared randomness

---

[4]Originally this definition was called reproducibility [Impagliazzo et al., 2022] but it was later pointed out that the correct term is replicability [Ahn et al., 2022].

between the two executions. In practice, it can be implemented as a shared random seed. We underline that sharing the randomness across executions is crucial for the development of our algorithms. We also note that by doing that we *couple* the two random variables $\mathcal{A}(\bar{X}), \mathcal{A}(\bar{Y})$, whose realization depends on $r \sim R(\mathcal{X})$. Thus, if their realizations are equal with high probability under this coupling, it means that the distributions of $\mathcal{A}(\bar{X}), \mathcal{A}(\bar{Y})$ are *statistically close*. This connection is discussed further in Kalavasis et al. [2023].

In the context of a clustering algorithm $\mathcal{A}$, we interpret the output $\mathcal{A}(\bar{X}; \bar{r})$ as a clustering function $f : \mathcal{X} \to [k]$ which partitions the support of $\mathbb{P}$. The definition of $\rho$-replicability demands that $f$ is the same with probability at least $1 - \rho$ across two executions. We note that in the case of centroid-based clustering such as $k$-medians and $k$-means, the induced partition is a function of the centers and thus it is sufficient to output the exact same centers with probability $1 - \rho$ across two executions. However, we also allow for algorithms that create partitions implicitly without computing their centers explicitly.

Our goal is to develop replicable clustering algorithms for $k$-medians, $k$-means, and $k$-centers, which necessitates that the centers we choose are arbitrary points within $\mathbb{R}^d$ and *not* only points among the samples. We underline that as in the case of differential privacy, it is trivial to design algorithms that satisfy the replicability property, e.g. we can let $\mathcal{A}$ be the constant mapping. The catch is that these algorithms do not achieve any *utility*. In this work, we are interested in designing replicable clustering algorithms whose utility is competitive with their non-replicable counterparts.

## 3 Main Results

In this section, we informally state our results for replicable statistical $k$-medians ($(k, 1)$-clustering), $k$-means ($(k, 2)$-clustering), and $k$-centers under general distances. Unfortunately, generality comes at the cost of exponential dependency on the dimension $d$. We also state our results for replicable statistical $k$-medians and $k$-means specifically under the Euclidean distance, which has a polynomial dependency on $d$. Two key ingredients is the uniform convergence of $(k, p)$-clustering costs (cf. Theorem C.9) as well as a data-oblivious dimensionality reduction technique for $(k, p)$-clustering in the distributional setting (cf. Theorem E.10). These results may be of independent interest.

We emphasize that the Euclidean $k$-median and $k$-means are the most studied and applied flavors of clustering, thus the sample complexity for the general case and the restriction to $p = 1, 2$ does not diminish the applicability of our approach.

The main bottleneck in reducing the sample complexity for general norms is the lack of a data-oblivious dimensionality reduction scheme. This bottleneck is not unique to replicability and such a scheme for general norms would be immediately useful for many distance-based problems including clustering. It may be possible to extend our results to general $(k, p)$-clustering beyond $p = 1, 2$. The main challenge is to develop an approximate triangle inequality for $p$-th powers of norms. Again, this limitation is not due to replicability but rather the technique of hierarchical grids. It is a limitation shared by Frahling and Sohler [2005].

Before stating our results, we reiterate that the support of our domain $\mathcal{X}$ is a subset of the unit-diameter $\kappa$-ball $\mathcal{B}_d$. In particular, we have that $\mathrm{OPT} \leq 1$.

**Theorem 3.1** (Informal). *Let $\varepsilon, \rho \in (0, 1)$. Given black-box access to a $\beta$-approximation oracle for weighted $k$-medians, respectively weighted $k$-means (cf. Problem 2.2), there is a $\rho$-replicable algorithm for statistical $k$-medians, respectively $k$-means (cf. Problem 2.1), such that with probability at least $0.99$, it outputs a $(1 + \varepsilon)\beta$-approximation. Moreover, the algorithm has sample complexity*

$$\tilde{O}\left(\mathrm{poly}\left(\frac{k}{\rho\,\mathrm{OPT}}\right)\left(\frac{2\Delta}{\varepsilon}\right)^{O(d)}\right).$$

When we are working in Euclidean space, we can get improved results for these problems.

**Theorem 3.2** (Informal). *Let $\rho \in (0, 1)$. Suppose we are provided with black-box access to a $\beta$-approximation oracle for weighted Euclidean $k$-medians ($k$-means). Then there is a $\rho$-replicable algorithm that partitions the input space so with probability at least $0.99$, the cost of the partition is*

*at most $O(\beta \, \text{OPT})$. Moreover, the algorithm has sample complexity*

$$\tilde{O}\left(\text{poly}\left(\frac{d}{\text{OPT}}\right)\left(\frac{k}{\rho}\right)^{O(\log\log(k/\rho))}\right).$$

We underline that in this setting we compute an *implicit* solution to Problem 2.1, since we do not output $k$ centers. Instead, we output a function $f$ that takes as input a point $x \in \mathcal{X}$ and outputs the label of the cluster it belongs to in polynomial time. The replicability guarantee states that, with probability $1 - \rho$, the function will be the same across two executions.

The combinatorial $k$-medians ($k$-means) problem where the centers are restricted to be points of the input is a well-studied problem from the perspective of polynomial-time constant-factor approximation algorithms. See Byrka et al. [2017], Ahmadian et al. [2019] for the current best polynomial-time approximation algorithms for combinatorial $k$-medians ($k$-means) in general metrics with approximation ratio 2.675 (9). Also, see Cohen-Addad et al. [2022] for a 2.406 (5.912) approximation algorithm for the combinatorial Euclidean $k$-medians ($k$-means).

As we alluded to before, in order to solve $k$-centers from samples, we need to make an additional assumption. Essentially, Assumption F.2 states that there is a $(\beta, B)$-approximate solution $F$, such that, with some constant probability, e.g. 0.99, when we draw $n$ samples from $\mathbb{P}$ we will observe at least one sample from each cluster of $F$.

**Theorem 3.3** (Informal). *Let $c \in (0, 1)$. Given black-box access to a $(\hat{\beta}, \hat{B})$-approximation oracle for $k$-centers (cf. Problem 2.4) and under Assumption F.2, there is a $\rho$-replicable algorithm for statistical $k$-centers (cf. Problem 2.3), that outputs a $(O(\beta + \hat{\beta}), O(B + \hat{B} + (\beta + \hat{\beta} + 1)c)\Delta)$-approximate solution with probability at least 0.99. Moreover, it has sample complexity*

$$\tilde{O}\left(\frac{n^2 k \, (1/c)^{3d}}{\rho^2 q^2}\right).$$

Recall that there is a simple greedy 2-approximation for the sample $k$-center problem whose approximation ratio cannot be improved unless P = NP [Hochbaum and Shmoys, 1985].

We defer the discussion around $k$-centers to Appendix F. In particular, see Theorem F.8 in Appendix F for the formal statement of Theorem 3.3.

## 4 Overview of $(k, p)$-Clustering

In this section, we present our approach to the $(k, p)$-clustering problem. First, we replicably approximate the distribution with a finite set of points by extending the approach of Frahling and Sohler [2005] to the distributional setting. Then, we solve the combinatorial $(k, p)$-clustering problem on this coreset using an approximation oracle in a black-box manner. In the following subsections, we give a more detailed overview for each step of our approach. For the full proofs and technical details, we kindly refer the reader to Appendix D.4 - D.6. In summary:

1) Replicably build a variant of a *quad tree* [Finkel and Bentley, 1974] (cf. Section 4.2).
2) Replicably produce a weighted *coreset* using the quad tree (cf. Section 4.1).
3) Apply the optimization oracle for the combinatorial problem on the coreset.

For general norms, this approach leads to an exponential dependence on $d$. However, we are able to handle the case of Euclidean distances by extending existing dimensionality reduction techniques for sample Euclidean $(k, p)$-clustering [Makarychev et al., 2019] to the distributional case (cf. Section 5). Thus, for the widely used Euclidean norm, our algorithm has $\text{poly}(d)$ sample complexity.

### 4.1 Coresets

**Definition 4.1** ((Strong) Coresets). For a distribution $\mathbb{P}$ with support $\mathcal{X} \subseteq \mathcal{B}_d$ and $\varepsilon \in (0, 1)$, a *(strong) $\varepsilon$-coreset* for $\mathcal{X}$ is a distribution $\mathbb{P}'$ on $\mathcal{X}' \subseteq B_d$ which satisfies

$$(1 - \varepsilon) \cdot \mathbb{E}_{x \sim \mathbb{P}} \kappa(x, F)^p \leq \mathbb{E}_{x' \sim \mathbb{P}'}\left[\kappa(x', F)^p\right] \leq (1 + \varepsilon) \cdot \mathbb{E}_{x \sim \mathbb{P}} \kappa(x, F)^p$$

for every set of centers $F \subseteq \mathcal{B}_d, |F| = k$.

Essentially, coresets help us approximate the true cost on the distribution $\mathbb{P}$ by considering another distribution $\mathbb{P}'$ whose support $\mathcal{X}'$ can be arbitrarily smaller than the support of $\mathbb{P}$.

Inspired by Frahling and Sohler [2005], the idea is to replicably consolidate our distribution $\mathbb{P}$ through some mapping $R : \mathcal{X} \to \mathcal{X}$ whose image has small cardinality $|R(\mathcal{X})| << \infty$ so that for any set of centers $F$,

$$(1 - \varepsilon)\mathbb{E}_x \kappa(x, F)^p \leq \mathbb{E}_x \kappa(R(x), F)^p \leq (1 + \varepsilon)\mathbb{E}_x \kappa(x, F)^p ,$$

where $\varepsilon \in (0, 1)$ is some error parameter. In other words, $(\mathbb{P}_R, R(\mathcal{X}))$ is an $\varepsilon$-coreset. Note that given the function $R$, we can replicably estimate the probability mass at each point in $R(\mathcal{X})$ and then apply a weighted $(k, p)$-clustering algorithm.

## 4.2 Replicable Quad Tree

We now explain how to replicably obtain the mapping $R : \mathcal{X} \to \mathcal{X}$ by building upon the work of Frahling and Sohler [2005]. The pseudocode of the approach is provided in Algorithm 4.1. While Frahling and Sohler [2005] present their algorithm using hierarchical grids, we take an alternative presentation using the quad tree [Finkel and Bentley, 1974], which could be of independent interest.

First, we recall the construction of a standard quad tree in dimension $d$. Suppose we have a set of $n$ points in $[-1/2, 1/2]^d$. The quad tree is a tree whose nodes represent hypercubes containing points and can be built recursively as follows: The root represents the cell $[-1/2, 1/2]^d$ and contains all points. If a node contains more than one point, we split the cell it represents into $2^d$ disjoint, equally sized cells. For each non-empty cell, we add it as a child of the current node and recurse into it. The recursion stops when a node contains only 1 point. In the distributional setting, the stopping criterion is either when the diameter of a node is less than some length or when the node contains less than some mass, where both quantities are a function of the depth of the node. See Algorithm 4.1.

A quad tree implicitly defines a function $R : \mathcal{X} \to \mathcal{X}$ as follows. Given a point $x \in \mathcal{X}$ and the root node of our tree, while the current node has a child, go to the child containing $x$ if such a child exists, otherwise, go to any child. At a leaf node, output the center of the cell the leaf represents. Intuitively, the quad tree consolidates regions of the sample space into single points. The construction can be made replicable since the decision to continue the recursion or not is the only statistical operation and is essentially a heavy-hitters operations which can be performed in a replicable fasion.

Let $\mathcal{G}_i$ denote the union of all $2^{id}$ possible cells at the $i$-th level. We write $\mathbb{P}_i$ to denote the discretized distribution to $\mathcal{G}_i$. In other words, $\mathbb{P}_i = \mathbb{P}|_{\sigma(\mathcal{G}_i)}$ is the restriction of $\mathbb{P}$ to the smallest $\sigma$-algebra containing $\mathcal{G}_i$. Moreover, we write $\Lambda$ to denote a replicable estimate of OPT with relative error $\varepsilon$, say $1/\beta(1+\varepsilon)$ OPT $\leq \Lambda \leq (1 + \varepsilon)$ OPT for some absolute constant $\beta \geq 1$. We demonstrate how to obtain such a replicable estimate in Appendix D.5.

---

**Algorithm 4.1** Replicable Quad Tree

---
1: **rQuadTree**(distribution $\mathbb{P}$, accuracy $\varepsilon$, exponent $p$, replicability $\rho$, confidence $\delta$):
2: Init the node on the first level $\mathcal{Z}[0] \leftarrow \left\{[-1/2, 1/2]^d\right\}$.
3: **for** depth $i \leftarrow 1$; $\mathcal{Z}[i - 1] \neq \varnothing$ AND $(2^{-i+1}\Delta)^p > \varepsilon\Lambda/5$; $i \leftarrow i + 1$ **do**
4: $\quad$ {$\mathbb{P}_i$ is the discretized distribution over $2^{id}$ cells on the $i$-th layer}
5: $\quad$ {$\gamma$ is a parameter to be determined later.}
6: $\quad$ {$t$ is an upper bound on the number of layers.}
7: $\quad$ Compute $\mathcal{H} = \text{rHeavyHitters}\left(\mathbb{P}_i, v = \frac{\gamma \cdot \Lambda}{2^{-pi}}, \frac{v}{2}, \frac{\rho}{t}, \frac{\delta}{t}\right)$
8: $\quad$ **for** node $Z \in \mathcal{Z}[i - 1]$ **do**
9: $\quad\quad$ **for** heavy hitter cells $H \in \mathcal{H}$ such that $H \subseteq Z$ **do**
10: $\quad\quad\quad$ children($Z$) $\leftarrow$ children($Z$) $\cup \{H\}$
11: $\quad\quad\quad$ $\mathcal{Z}[i] \leftarrow \mathcal{Z}[i] \cup \{H\}$.
12: $\quad\quad$ **end for**
13: $\quad$ **end for**
14: **end for**
15: Output root node.

---

Our technique differs from that of Frahling and Sohler [2005] in at least three ways. Firstly, they performed their analysis for finite, uniform distributions with access to the entire distribution, while

our results hold assuming only sample access to a general bounded distribution[5]. Secondly, Frahling and Sohler [2005] bound the number of layers as a function of the cardinality of the support. For us, this necessitates the extra termination condition when the side lengths of our grids fall below a fraction of $\Lambda$ as our distribution may have infinite support. Finally, Frahling and Sohler [2005] estimate OPT by enumerating powers of 2. This suffices for their setting since their distributions are discrete and bounded. However, we require a more nuanced approach (cf. Appendix D.5) as we do not have a lower bound for OPT. We tackle this by showing uniform convergence of the clustering solution costs, which we establish via metric entropy and Rademacher complexity (cf. Appendix C).

## 4.3 Putting it Together

Once we have produced the function $R$ implicitly through a quad tree, there is still the matter of extracting a replicable solution from a finite distribution. We can accomplish this by replicably estimating the probability mass at each point of $R(\mathcal{X})$ and solving an instance of the weighted sample $k$-medians ($k$-means). This leads to the following results. For details, see Appendix D.4 - D.6.

**Theorem 4.2** (Theorem 3.1; Formal). *Let $\varepsilon, \rho \in (0,1)$ and $\delta \in (0, \rho/3)$. Given black-box access to a $\beta$-approximation oracle for weighted $k$-medians (cf. Problem 2.2), there is a $\rho$-replicable algorithm for statistical $k$-medians (cf. Problem 2.1) such that, with probability at least $1 - \delta$, it outputs a $(1 + \varepsilon)\beta$-approximation. Moreover, it has sample complexity*

$$\tilde{O}\left(\left(\frac{k^2 d^2}{\varepsilon^{12}\rho^6 \cdot \mathrm{OPT}^{12}} + \frac{k^3 2^{18d}\Delta^{3d+3}}{\rho^2\varepsilon^{3d+5} \cdot \mathrm{OPT}^3}\right)\log\frac{1}{\delta}\right).$$

**Theorem 4.3** (Theorem 3.1; Formal). *Given black-box access to a $\beta$-approximation oracle for weighted $k$-means (cf. Problem 2.2), there is a $\rho$-replicable algorithm for statistical $k$-means (cf. Problem 2.1) such that, with probability at least $1 - \delta$, it replicably outputs a $(1+\varepsilon)\beta$-approximation. Moreover, it has sample complexity*

$$\tilde{O}\left(\left(\frac{k^2 d^2}{\varepsilon^{12}\rho^6 \cdot \mathrm{OPT}^{12}} + \frac{k^3 2^{39d}\Delta^{3d+6}}{\rho^2\varepsilon^{6d+8} \cdot \mathrm{OPT}^3}\right)\log\frac{1}{\delta}\right).$$

## 5 The Euclidean Metric, Dimensionality Reduction, and $(k,p)$-Clustering

In this section, we focus on the Euclidean metric $\kappa(y,z) := \|y - z\|_2$ and show how the sample complexity for Euclidean $(k,p)$-clustering can be made polynomial in the ambient dimension $d$. Note that results for dimensionality reduction exist for the combinatorial Euclidean $(k,p)$-clustering problem [Charikar and Waingarten, 2022, Makarychev et al., 2019]. However, these do not extend trivially to the distributional case. We remark that we require our dimensionality reduction maps to be *data-oblivious* since we are constrained by replicability requirements.

The cornerstone result in dimensionality reduction for Euclidean distance is the Johnson-Lindenstrauss lemma (cf. Theorem E.1), which states that there is a distribution over linear maps $\pi_{d,m}$ from $\mathbb{R}^d \to \mathbb{R}^m$ that approximately preserves the norm of any $x \in \mathbb{R}^d$ with constant probability for some target dimension $m$. In Makarychev et al. [2019], the authors prove Theorem E.4, which roughly states that it suffices to take $m = \tilde{O}(p^4/\varepsilon^2 \log(1/\delta))$ in order to preserve $(k,p)$-clustering costs when the domain is finite. Firstly, we extend Theorem E.4 to the distributional setting by implicitly approximating the distribution with a weighted $\varepsilon$-net. Then, we implicitly map the $\varepsilon$-net onto the low-dimensional space and solve the clustering problem there. An important complication we need to overcome is that this mapping preserves the costs that correspond to *partitions*[6] of the data and not arbitrary solutions. Because of that, it is not clear how we can "lift" the solution from the low-dimensional space to the original space. Thus, instead of outputting $k$ points that correspond to the centers of the clusters, our algorithm outputs a *clustering function* $f : \mathcal{X} \to [k]$, which takes as input a point $x \in \mathcal{X}$ and returns the label of the cluster it belongs to. The replicability guarantees of the algorithm state that, with probability $1 - \rho$, it will output the *same* function across two executions. In Appendix E.4, we describe this function. We emphasize that for each $x \in \mathcal{X}$, the running time of $f$ is polynomial in $k, d, p, 1/\varepsilon, \log(1/\delta)$. Essentially, this function maps $x$ onto the low-dimensional

---

[5]Our results also generalize to unbounded distributions with sufficiently small tails

[6]Roughly speaking, a solution corresponds to a partition when the center of each cluster is its center of mass.

space using the same projection map $\pi$ as our algorithm and then finds the nearest center of $\pi(x)$ in the low-dimensional space. For full details, we refer the reader to Appendix E.3. We are now ready to state the result formally.

**Theorem 5.1** (Theorem 3.2; Formal). *Let $\varepsilon, \rho \in (0,1)$ and $\delta \in (0, \rho/3)$. Given a $\beta$-approximation oracle for weighted Euclidean $k$-medians ($k$-means), there is a $\rho$-replicable algorithm that outputs a clustering function such that with probability at least $1 - \delta$, the cost of the partition is at most $(1 + \varepsilon)\beta \, \mathrm{OPT}$. Moreover, the algorithm has sample complexity*

$$\tilde{O} \left( \mathrm{poly} \left( \frac{kd}{\rho \, \mathrm{OPT}} \right) \left( \frac{2\sqrt{m}}{\varepsilon} \right)^{O(m)} \log \frac{1}{\delta} \right),$$

*where $m = O\left( \frac{1}{\varepsilon^2} \log \frac{k}{\delta \varepsilon} \right)$.*

## 6 Running Time for $(k, p)$-Clustering

All of our algorithms terminate in $O(\mathrm{poly}(n))$ time where $n$ denotes the sample complexity. See Table A.1 for more detailed time complexity of each stage of our approach. Moreover, we make $O(\log(\beta/(\varepsilon \, \mathrm{OPT})))$ calls to the $\beta$-approximation oracle for the combinatorial $(k, p)$-clustering problem within our OPT estimation subroutine and then one more call to the oracle in order to output a solution on the coreset.

## 7 Replicable $k$-Centers

Due to space limitations, we briefly sketch our approach for the $k$-centers problem and kindly refer the reader to Appendix F. As explained before, the assumptions we make in this setting state that there exists some "good" solution, so that when we draw $n$ i.i.d. samples from $\mathbb{P}$ we observe at least one sample from each cluster, with constant probability. We first take a fixed grid of side $c$ in order to cover the unit-diameter ball. Then, we sample sufficiently many points from $\mathbb{P}$. Subsequently, we "round" all the points of the sample to the centers of the cells of the grid that they fall into and estimate the probability mass of every cell. In order to ensure replicability, we take a random threshold from a predefined interval and discard the points from all the cells whose mass falls below the threshold. Finally, we call the approximation oracle using the points that remain. Unlike the $(k, p)$-clustering problem (cf. Problem 2.1), to the best of our knowledge, there does not exist any dimensionality reduction techniques that apply to the $k$-centers problem. The main result is formally stated in Theorem F.8.

## 8 Experiments

We now provide a practical demonstration[7] of the replicability of our approach on synthetic data in 2D. In Figure 8.1, we leverage the sklearn [Pedregosa et al., 2011] implementation of the popular $k$-means++ algorithm for $k = 3$ and compare the output across two executions on the two moons distribution. In the first experiment, we do not perform any preprocessing and run $k$-means++ as is, resulting in different centers across two executions. In the second experiment, we compute a replicable coreset for the two moons distribution before running $k$-means++ on the coreset. This leads to the same centers being outputted across the executions. Note that the computation for the coreset is performed independently for each execution, albeit with a shared random seed for the internal randomness. See also Figure 8.2 for the results of a similar experiment on a mixture of truncated Gaussian distributions.

---

[7]https://anonymous.4open.science/r/replicable_clustering_experiments-E380

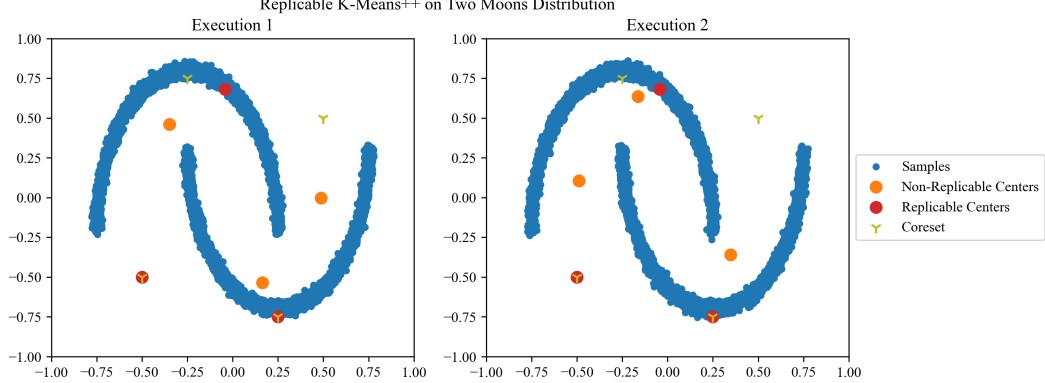

Figure 8.1: The results of running vanilla vs replicable $k$-Means++ on the two moons distribution for $k = 3$.

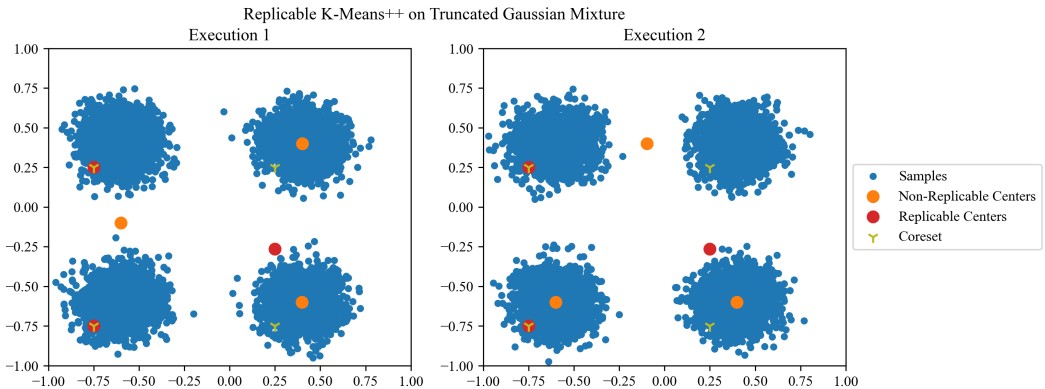

Figure 8.2: The results of running vanilla vs replicable $k$-Means++ on a mixture of truncated Gaussians distributions for $k = 3$.

## 9  Conclusion & Future Work

In this work, we designed replicable algorithms with strong performance guarantees using black-box access to approximation oracles for their combinatorial counterparts. There are many follow-up research directions that this work can lead to. For instance, our coreset algorithm adapts the coreset algorithm of Frahling and Sohler [2005] by viewing their algorithm as a series of heavy hitter estimations that can be made replicable. it may be possible to interpret more recent approaches for coreset estimation as a series of statistical operations to be made replicable in order to get replicable algorithms in the statistical $(k, p)$-clustering setting [Hu et al., 2018]. It would also be interesting to examine the sensitivity of (replicable) clustering algorithms to the choice of parameters such as the choice of exponent in the cost function or the measure of similarity of the data. Another relevant direction is to explore sample complexity lower ounds for statistical clustering, where little is known even in the non-replicable setting.

## Acknowledgments and Disclosure of Funding

Amin Karbasi acknowledges funding in direct support of this work from NSF (IIS-1845032), ONR (N00014-19-1-2406), and the AI Institute for Learning-Enabled Optimization at Scale (TILOS). Grigoris Velegkas is supported by TILOS, the Onassis Foundation, and the Bodossaki Foundation. Felix Zhou is supported by TILOS.

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

## A  Tables

We present the running times required for our sub-routines in order to ensure each stage is $\rho$-replicable and succeeds with probability at least 0.99. The overall output is a constant ratio approximation for Euclidean $k$-medians ($k$-means).

Table A.1: Running Time Overview for Euclidean $k$-Medians ($k$-Means)

| ALGORITHM | RUNNING TIME |
|---|---|
| OPT ESTIMATION | $\tilde{O}(\text{poly}(kd/\text{OPT }\rho))$ |
| DIMENSIONALITY REDUCTION | $\tilde{O}(\text{poly}(d/\text{OPT})(k/\rho)^{O(\log\log(k/\rho))})$ |
| CORESET | $\tilde{O}(\text{poly}(d/\text{OPT})(k/\rho)^{O(\log\log(k/\rho))})$ |
| PROBABILITY MASS ESTIMATION | $\tilde{O}(\text{poly}(d/\text{OPT})(k/\rho)^{O(\log\log(k/\rho))})$ |

## B  Useful Facts

**Proposition B.1** (Bretagnolle-Huber-Carol Inequality; [Vaart and Wellner, 1997]). *Suppose the random vector* $(Z^{(1)}, \ldots, Z^{(N)})$ *is multinomially distributed with parameters* $(p^{(1)}, \ldots, p^{(N)})$ *and* $n$. *Let* $\widehat{p}^{(j)} := \frac{1}{n}Z^{(j)}$. *Then*

$$\mathbb{P}\left\{\sum_{j=1}^{N}|\widehat{p}^{(j)} - p^{(j)}| \geq 2\varepsilon\right\} \leq 2^N \exp\left(-2\varepsilon^2 n\right).$$

*In particular, for any* $\varepsilon, \rho \in (0,1)$*, sampling*

$$n \geq \frac{\ln\frac{1}{\rho} + N\ln 2}{2\varepsilon^2}$$

*points from a finite distribution implies that* $\sum_{j=1}^{N}|\widehat{p}^{(j)} - p^{(j)}| < 2\varepsilon$ *with probability at least* $1 - \rho$.

*Remark* B.2 (Weighted $k$-Means/Medians). We remark that if we have access to a $\beta$-approximation oracle for unweighted $k$-means/medians, we can implement a weighted one by considering multiple copies of the points. In particular, in our applications, we get weights that are polynomials in the parameters of concern so this will not affect the stated runtime guarantees.

## C  Uniform Convergence of $(k, p)$-Clustering Costs

We would like to estimate OPT for statistical $k$-medians and statistical $k$-means by solving the combinatorial problem on a sufficiently large sample size. However, while the convergence of cost for a particular set of centers is guaranteed by standard arguments, e.g. Chernoff bounds, we require the stronger statement that convergence holds simultaneously for *all* possible choices of centers.

Similar results can be found [Ben-David, 2007], with the limitation that centers are either chosen from the sample points, or chosen as centers of mass of the clusters they induce. While this suffices to achieve constant ratio approximations, we would like to choose our centers anywhere in $\mathcal{B}_d$. One reason for doing so is for replicability as we have no control over the location of samples from two independent executions so we need to output centers that are not overly reliant on its specific input. Another is for dimensionality reduction, which we will see later.

### C.1  History of Uniform Laws of Large Numbers

Uniform laws of large numbers generalize convergence results for a finite number of random variables to possibly uncountable classes of random variables. The Rademacher complexity and related Gaussian complexity are canonical techniques in developing these laws and also have a lengthy history in the study of Banach spaces using probabilistic methods [Pisier, 1999, Milman and Schechtman,

1986, Ledoux and Talagrand, 1991]. Metric entropy, along with related notions of expressivity of various function classes, can be used to control the Rademacher complexity and are also central objects of study in the field of approximation theory [DeVore and Lorentz, 1993, Carl and Stephani, 1990].

## C.2 Rademacher Complexity

Let $\mathscr{F}$ denote a class of functions from $\mathbb{R}^d \to \mathbb{R}$. For any fixed collection of $n$ points $x_1^n := (x_1, \ldots, x_n)$, we write

$$f(x_1^n) := (f(x_1), \ldots, f(x_n))$$
$$\mathscr{F}(x_1^n) := \{f(x_1^n) : f \in \mathscr{F}\},$$

i.e, the restriction of $\mathscr{F}$ onto the sample. Let $s \in \{-1, 1\}^n$ be a Rademacher random vector. That is, $s_i$ takes on values $-1, 1$ with equal probability independently of other components. Recall that the *Rademacher complexity* of $\mathscr{F}$ is given by

$$\mathcal{R}_n(\mathscr{F}) := \mathbb{E}_{X,s}\left[\sup_{f \in \mathscr{F}} \frac{1}{n}\langle s, f(X_1^n)\rangle\right].$$

**Theorem C.1** ([Wainwright, 2019]). *For any b-uniformly bounded class of functions $\mathscr{F}$, integer $n \geq 1$, and error $\varepsilon \geq 0$,*

$$\sup_{f \in \mathscr{F}}\left|\frac{1}{n}\sum_{i=1}^{n} f(X_i) - \mathbb{E}[f(X)]\right| \leq 2\mathcal{R}_n(\mathscr{F}) + \varepsilon$$

*with probability at least $1 - \exp\left(-\frac{n\varepsilon^2}{2b^2}\right)$.*

*In particular, as long as $\mathcal{R}_n(\mathscr{F}) = o(1)$, we have uniform convergence of the sample mean.*

## C.3 Metric Entropy

We write $B(x, r; \mu)$ to denote the ball of radius $r$ about a point $x$ with respect to a metric $\mu$. Recall that an $\varepsilon$-*cover* of a set $T$ with respect to a metric $\mu$ is a subset $\theta_1, \ldots, \theta_N \subseteq T$ such that

$$T \subseteq \bigcup_{i \in [N]} B(\theta_i, \varepsilon; \mu).$$

The *covering number* $N(\varepsilon; T, \mu)$ is the cardinality of the smallest $\varepsilon$-cover.

**Proposition C.2** ([Wainwright, 2019]). *Fix $\varepsilon \in (0, 1)$. Let $B(R; \|\cdot\|)$ denote the d-dimensional ball of radius R with respect to $\|\cdot\|$. Then*

$$N(\varepsilon, B(R; \|\cdot\|), \|\cdot\|) \leq \left(1 + \frac{2R}{\varepsilon}\right)^d \leq \left(\frac{3R}{\varepsilon}\right)^d.$$

We say a collection of zero-mean random variables $\{Y_\theta : \theta \in T\}$ is a *sub-Gaussian process* with respect to a metric $\mu$ on $T$ if

$$\mathbb{E}\exp\left[\lambda(Y_\theta - Y_{\theta'})\right] \leq \exp\left[\frac{\lambda^2 \mu^2(\theta, \theta')}{2}\right]$$

for all $\theta, \theta' \in T$ and $\lambda \in \mathbb{R}$.

**Proposition C.3.** *The canonical Rademacher process*

$$Y_\theta := \langle s, \theta\rangle$$

*is a zero-mean sub-Gaussian process with respect to the Euclidean norm on $T \subseteq \mathbb{R}^n$.*

*Proof.* Recall that a Rademacher variable is sub-Gaussian with parameter 1. Moreover, the sum of sub-Gaussian variables is sub-Gaussian with parameter equal to the Euclidean norm of the parameters. It follows that $Y_\theta - Y_{\theta'} = \langle s, \theta - \theta'\rangle$ is sub-Gaussian with parameter $\|\theta - \theta'\|_2$. The result follows. $\qquad\square$

**Theorem C.4** (One-Step Discretization; [Wainwright, 2019]). *Let $Y_\theta, \theta \in T$ be a zero-mean sub-Gaussian process with respect to the metric $\mu$. Fix any $\varepsilon \in [0, D]$ where $D = \mathrm{diam}(T; \mu)$ such that $N(\varepsilon; T, \mu) \geq 10$. Then*

$$\mathbb{E}\left[\sup_{\theta \in T} Y_\theta\right] \leq \mathbb{E}\left[\sup_{\theta, \theta' \in T} Y_\theta - Y_{\theta'}\right]$$

$$\leq 2\mathbb{E}\left[\sup_{\theta, \theta' \in T: \mu(\theta, \theta') \leq \varepsilon} Y_\theta - Y_{\theta'}\right] + 4D\sqrt{\log N(\varepsilon; T, \mu)}.$$

In general, it may be possible to improve the bound in Theorem C.4 using more sophisticated techniques such as Dudley's entropy integral bound [Wainwright, 2019]. However, the simple inequality from Theorem C.4 suffices for our setting.

**Corollary C.5.** *Fix any $\varepsilon \in [0, D]$ where $D = \mathrm{diam}(T; \mu)$ such that $N(\varepsilon; T, \mu) \geq 10$. The canonical Rademacher process $Y_\theta = \langle s, \theta \rangle$ for $\theta \in T \subseteq \mathbb{R}^n$ satisfies*

$$\mathbb{E}\left[\sup_\theta Y_\theta\right] \leq 2\varepsilon\sqrt{n} + 4D\sqrt{\log N(\varepsilon; T, \|\cdot\|_2)}.$$

*Proof.* By Proposition C.3, $Y_\theta$ is a zero-mean sub-Gaussian process and we can apply Theorem C.4 to conclude that

$$\mathbb{E}\left[\sup_{\|\theta - \theta'\|_2 \leq \varepsilon} Y_\theta - Y_\theta'\right] = 2\mathbb{E}\left[\sup_{\|v\|_2 \leq \varepsilon} \langle v, s \rangle\right]$$

$$\leq 2\mathbb{E}\left[\|s\|_2 \cdot \|v\|_2\right]$$

$$\leq 2\varepsilon\sqrt{n}. \qquad \square$$

Corollary C.5 gives us a way to control the Rademacher complexity of a function class whose co-domain is well-behaved. We make this notion rigorous in the next section.

### C.4 Uniform Convergence of $(k, p)$-Clustering Cost

For a fixed set of centers $F$, let $\kappa_F^p : \mathbb{R}^d \to \mathbb{R}$ be given by

$$\kappa_F^p(x) := \kappa(x, F)^p.$$

Define $\mathcal{F} := \{F \subseteq \mathcal{B}_d : |F| = k\}$. We take our function class to be

$$\mathscr{F} := \{\kappa_F^p : F \in \mathcal{F}\}.$$

Let $\mu : \mathcal{F} \times \mathcal{F} \to \mathbb{R}$ be given by

$$\mu(A, B) := \max_{a \in A, b \in B} \kappa(a, b).$$

**Proposition C.6.** *$\mu$ is a metric on $\mathcal{F}$.*

*Proof.* It is clear that $\mu$ is symmetric and positive definite. We need only show that the triangle inequality holds. Fix $A, B, C \in \mathcal{F}$ and suppose $a \in A, c \in C$ are such that $\kappa(a, c) = \mu(A, C)$. Then

$$\mu(A, C) = \kappa(a, c)$$
$$\leq \kappa(a, b) + \kappa(b, c) \qquad \forall b \in B$$
$$\leq \mu(A, B) + \mu(B, C)$$

as desired. $\qquad \square$

**Proposition C.7.** *$\mathscr{F}$ is $p$-Lipschitz parameterized with respect to the metric $\mu$.*

*Proof.* We have

$$\begin{aligned}
|\kappa_F^1(x) - \kappa_{F'}^1(x)| &= |\kappa(x, F(x)) - \kappa(x, F'(x))| \\
&\leq |\kappa(x, F'(x)) + \kappa(F'(x), F(x)) - \kappa(x, F'(x))| \\
&= \kappa(F'(x), F(x)) \\
&\leq \mu(F, F').
\end{aligned}$$

Now, the function $g(x) : [0,1] \to \mathbb{R}$ given by $x \mapsto x^p$ is $p$-Lipschitz by the mean value theorem:

$$|g(x) - g(y)| \leq \sup_{\xi \in [0,1]} g'(\xi)|x - y| \leq p|x - y|.$$

The result follows by the fact that the composition of Lipschitz functions is Lipschitz with a constant equal to the product of constants from the composed functions. $\square$

**Theorem C.8.** *For any $\varepsilon \in [0, D = \sqrt{n}]$ such that $N(\varepsilon; \mathscr{F}(x_1^n), \|\cdot\|_2) \geq 10$,*

$$\mathcal{R}_n(\mathscr{F}) \leq \frac{2\varepsilon}{\sqrt{n}} + 4\sqrt{\frac{kd \log \frac{3\sqrt{n}p}{\varepsilon}}{n}} = o(1).$$

*Proof.* Remark that $\kappa_F^p(x_1^n)$ is $\sqrt{n}p$-Lipschitz with respect to the metric $\mu$. Indeed,

$$\sum_{i=1}^n |\kappa_F^p(x_i) - \kappa_{F'}^p(x_i)|^2 \leq np^2 \mu(F, F')^2.$$

Now, $D := \operatorname{diam}(\mathscr{F}(x_1^n); \|\cdot\|_2) = \sqrt{n}$. We can apply Corollary C.5 with $T = \mathscr{F}(x_1^n)$ to see that

$$\begin{aligned}
\mathcal{R}_n(\mathscr{F}) &= \frac{1}{n} \mathbb{E}\left[\sup_\theta Y_\theta\right] \\
&\leq \frac{2\varepsilon}{\sqrt{n}} + 4\sqrt{\frac{\log N(\varepsilon; \mathscr{F}(x_1^n), \|\cdot\|_2)}{n}}.
\end{aligned}$$

Now, since $\kappa_F^p(x_1^n)$ is $\sqrt{n}p$-Lipschitz, a $\frac{\varepsilon}{\sqrt{n}p}$-cover for $\mathcal{F}$ yields an $\varepsilon$-cover of $\mathscr{F}(x_1^n)$. Hence

$$N(\varepsilon; \mathscr{F}(x_1^n), \|\cdot\|_2) \leq N\left(\frac{\varepsilon}{\sqrt{n}p}; \mathcal{F}, \mu\right).$$

Note that the cross product of $k$ $\varepsilon$-covers of $\mathcal{B}$ is an $\varepsilon$-cover of $\mathcal{F}$. Hence

$$\begin{aligned}
N\left(\frac{\varepsilon}{\sqrt{n}p}; \mathcal{F}, \mu\right) &\leq N\left(\frac{\varepsilon}{\sqrt{n}p}; \mathcal{B}_d, \kappa\right)^k \\
&\leq \left(\frac{3\sqrt{n}p}{\varepsilon}\right)^{kd}. \qquad \text{Proposition C.2}
\end{aligned}$$

Substituting this bound on the covering number of $\mathscr{F}(x_1^n)$ concludes the proof. $\square$

Note that the Lipschitz property was crucial to ensure that the exponent in the covering number does not contain $n$.

**Theorem C.9.** *Fix $\varepsilon, \delta \in (0, 1)$. Then with*

$$O\left(\frac{k^2 d^2}{\varepsilon^4} \log \frac{p}{\delta}\right)$$

*i.i.d. samples from $\mathbb{P}$,*

$$\left|\widehat{\operatorname{cost}}(F) - \operatorname{cost}(F)\right| \leq \varepsilon$$

*with probability at least $1 - \delta$ for any set of centers $F \subseteq \mathcal{B}_d$.*

*Proof.* Choose $\varepsilon := 3 \le D = \sqrt{n}$. By Theorem C.8,

$$\mathcal{R}_n(\mathscr{F})$$

$$\le \frac{2 \cdot 3}{\sqrt{n}} + 4\sqrt{\frac{kd \log \frac{3\sqrt{np}}{3}}{n}}$$

$$\le \frac{6}{\sqrt{n}} + 4\sqrt{\frac{kd \log \sqrt{np}}{n}}$$

$$\le \frac{6}{\sqrt{n}} + 4\sqrt{\frac{kd \log p}{n}} + 4\sqrt{\frac{kd \log \sqrt{n}}{n}} \qquad\qquad \sqrt{a+b} \le \sqrt{a} + \sqrt{b}$$

$$\le \frac{6}{\sqrt{n}} + 4\sqrt{\frac{kd \log p}{n}} + 4\sqrt{\frac{kd}{\sqrt{n}}}. \qquad\qquad \log \sqrt{n} \le \sqrt{n}$$

Fix some $\varepsilon_1 \in (0,1)$. We have

$$\frac{6}{\sqrt{n}} \le \varepsilon_1$$

$$\iff n \ge \frac{36}{\varepsilon_1^2}.$$

Similarly, we have

$$4\sqrt{\frac{kd \log p}{n}} \le \varepsilon_1$$

$$\iff n \ge \frac{16kd \log p}{\varepsilon_1^2}.$$

Finally,

$$4\sqrt{\frac{kd}{\sqrt{n}}} \le \varepsilon_1$$

$$\iff n \ge \frac{256k^2d^2}{\varepsilon_1^4}.$$

By taking the maximum of the three lower bounds, we conclude that

$$n \ge \frac{256k^2d^2}{\varepsilon_1^4} \log p \implies \mathcal{R}_n(\mathscr{F}) \le 3\varepsilon_1.$$

Fix $\delta \in (0,1)$. Observe that $\mathscr{F}$ is 1-uniformly bounded. Thus by Theorem C.1, we require

$$\max\left(\frac{2}{\varepsilon_1^2} \log \frac{1}{\delta}, \frac{256k^2d^2}{\varepsilon_1^4} \log p\right) \le \frac{256k^2d^2}{\varepsilon_1^4} \log \frac{p}{\delta}$$

samples in order to guarantee that

$$\sup_{\kappa_F^p \in \mathscr{F}} \left| \frac{1}{n} \sum_{i=1}^n \kappa_F^p(X_i) - \mathbb{E}\kappa_F^p(X) \right| \le 2\mathcal{R}_n(\mathscr{F}) + \varepsilon_1$$

$$\le 7\varepsilon_1$$

with probability at least $1 - \delta$.

Choosing $\varepsilon_1 = \frac{\varepsilon}{7}$ concludes the proof. $\qquad\qquad\qquad\qquad\qquad\qquad\qquad\qquad \square$

# D $(k, p)$-Clustering

In this section, we provide the full proofs for Theorem 4.2 and Theorem 4.3, which state the guarantees for the replicable $k$-medians and $k$-means algorithms, respectively. First, we describe a

useful subroutine for replicable heavy hitters estimation in Appendix D.2. This subroutine is crucial to the replicable coreset algorithm (cf. Algorithm D.3), which we analyze in Appendix D.3. Once we have a coreset, it remains to solve the statistical $(k,p)$-clustering problem on a finite distribution. We describe a replicable algorithm for this in Appendix D.4. Our coreset algorithm assumes the knowledge of some constant ratio estimate of OPT. In Appendix D.5, we show how to output such an estimate replicably. Finally, we summarize our findings in Appendix D.6.

## D.1 Warm-Up: Replicable SQ Oracle and $\varepsilon$-Covers

In order to give some intuition to the reader, we first show how we can use a subroutine that was developed in Impagliazzo et al. [2022] in order to derive some results in the setting we are studying. We first need to define the *statistical query* model that was introduced in Kearns [1998]

**Definition D.1** (Statistical Query Oracle; [Kearns, 1998])**.** Let $\mathcal{D}$ be a distribution over the domain $\mathcal{X}$ and $\phi : \mathcal{X}^n \to \mathbb{R}$ be a statistical query with true value

$$v^\star := \lim_{n \to \infty} \phi(X_1, \dots, X_n) \in \mathbb{R}.$$

Here $X_i \sim_{i.i.d.} \mathcal{D}$ and the convergence is understood in probability or distribution. Let $\varepsilon, \delta \in (0,1)^2$. A *statistical query (SQ) oracle* outputs a value $v$ such that $|v - v^\star| \le \varepsilon$ with probability at least $1 - \delta$.

The simplest example of a statistical query is the sample mean

$$\phi(X_1, \dots, X_n) = \frac{1}{n} \sum_{i=1}^n X_i.$$

The idea is that by using a sufficiently large number of samples, an SQ oracle returns an estimate of the expected value of a statistical query whose range is bounded. Impagliazzo et al. [2022] provide a replicable implementation of an SQ oracle with a mild blow-up in the sample complexity which we state below.

**Theorem D.2** (Replicable SQ Oracle; [Impagliazzo et al., 2022])**.** *Let $\varepsilon, \rho \in (0,1)$ and $\delta \in (0, \rho/3)$. Suppose $\phi$ is a statistical query with co-domain $[0,1]$. There is a $\rho$-replicable SQ oracle to estimate its true value with tolerance $\varepsilon$ and failure rate $\delta$. Moreover, the oracle has sample complexity*

$$\tilde{O}\left(\frac{1}{\varepsilon^2 \rho^2} \log \frac{1}{\delta}\right).$$

The interpretation of the previous theorem is that we can replicably estimate statistical queries whose range is bounded.

We now explain how we can use this result for the statistical $k$-means problem under the Euclidean metric, but it is not hard to extend the approach to more general $(k,p)$-clustering problems. Consider a *fixed* set of centers $F$. Then, we can replicably estimate the cost of this solution using Theorem D.2 within an additive accuracy $\varepsilon$ and confidence $\delta$ using $O\left(\varepsilon^{-2} \rho^{-2} \log(1/\delta)\right)$ samples. Thus, a natural approach is to consider an $\varepsilon$-cover of the unit-diameter ball and then exhaustively search among solutions whose centers coincide with elements of the $\varepsilon$-cover. This is outlined in Algorithm D.1. We are now ready to state our results.

---

**Algorithm D.1** Replicable $k$-Means with $\varepsilon$-Cover

    **rKMeansCover:** distribution $\mathbb{P}$, error $\varepsilon$, replicability $\rho$, confidence $\delta$
    $G \leftarrow \varepsilon/3$-cover over the $d$-dimensional unit-diameter ball
    $\mathcal{F} \leftarrow \{F \subseteq G : |F| = k\}$ {We restrict the solutions to the $\varepsilon$-cover.}
    Output $\mathrm{argmin}_{F \in \mathcal{F}} \, \mathrm{rSQ\text{-}Oracle}(\mathrm{cost}(F), \varepsilon/3, \rho/|\mathcal{F}|, \delta/|\mathcal{F}|)$

---

**Lemma D.3.** *For any $\varepsilon, \delta, \rho \in (0,1), \delta < \rho/3$, Algorithm D.1 is $\rho$-replicable and outputs a solution $F$ whose cost is at most $\mathrm{OPT} + \varepsilon$ with probability $1 - \delta$. Moreover, it has sample complexity*

$$\tilde{O}\left(\frac{(9/\varepsilon)^{2kd}}{\rho^2 \varepsilon^2} \log \frac{1}{\delta}\right).$$

*Proof (Lemma D.3).* We first argue about the replicability of Algorithm D.1. Since we make $|\mathcal{F}|$ calls to the replicable SQ subroutine with parameter $\rho/|\mathcal{F}|$, the overall replicability of the algorithm follows by taking a union bound.

Let us now focus on the correctness of the algorithm. Let $F^*$ be the optimal solution. Consider the solution that we get when we move the centers of $F^*$ to the closest point of $G$ and let us denote it by $\hat{F}^*$. Notice that the cost of $\hat{F}^*$ is at most $\mathrm{OPT} + \varepsilon/3$. Furthermore, by a union bound, all the calls to the SQ oracle will return an estimate that is within an additive $\varepsilon/3$-error of the true cost. This happens with probability at least $1 - \delta$ and we condition on this event for the rest of the proof. Thus the estimated cost of the solution $\hat{F}^*$ will be at most $\mathrm{OPT} + 2\varepsilon/3$. Let $\widetilde{F}$ be the solution that we output. Its estimated cost is at most that of $\hat{F}^*$ and so its true cost will be at most $\mathrm{OPT} + \varepsilon$. This concludes the proof of correctness.

Lastly, we argue about the sample complexity of the algorithm. By Proposition C.2,

$$|\mathcal{F}| \leq O\left((9/\varepsilon)^{kd}\right).$$

By plugging this value into the sample complexity from Theorem D.2, we get

$$O\left(\frac{(9/\varepsilon)^{2kd}}{\rho^2 \varepsilon^2} \log \frac{(9/\varepsilon)^{kd}}{\delta}\right). \qquad \square$$

Although Algorithm D.1 provides some basic guarantees, there are several caveats with this approach. Namely, we can only get additive approximations and the dependence of the sample and time complexity on both $k, d$ is exponential.

In the following sections, we will explain how we can overcome these issues. As we alluded to before, our approach combines ideas from coresets estimation through hierarchical grids [Frahling and Sohler, 2005], uniform convergence through metric entropy [Wainwright, 2019], and dimensionality reduction techniques [Makarychev et al., 2019].

## D.2 Replicable Heavy-Hitters

In this section, we present a replicable heavy hitters algorithm which is inspired by Impagliazzo et al. [2022] and has an improved sample complexity by a factor of $O(1/(v-\varepsilon))$. We believe that this result could be of independent interest since the replicable heavy hitters algorithm has many applications as a subroutine in more complicated algorithms [Impagliazzo et al., 2022]. Intuitively, this algorithm consists of two phases. In the first phase, we estimate all the candidate heavy hitters and reduce the size of the domain. In the second phase, we estimate the mass of these candidates. We present a new analysis of the second phase using Proposition B.1.

---

**Algorithm D.2** Replicable Heavy Hitters

    **rHeavyHitters:** distribution $\mathbb{P}$, target $v$, error $\varepsilon$, replicability $\rho$, confidence $\delta$

**if** $|\mathcal{X}| < \frac{\ln \frac{2}{\delta(v-\varepsilon)}}{v-\varepsilon}$ **then**

    $\hat{\mathcal{X}} \leftarrow \mathcal{X}$

**else**

    $\hat{\mathcal{X}} \leftarrow \left\{\frac{\ln \frac{2}{\delta(v-\varepsilon)}}{v-\varepsilon} \text{ samples} \sim \mathbb{P}\right\}$

**end if**

$S \leftarrow \left\{\frac{648 \ln 2/\delta + 648\left(|\hat{\mathcal{X}}|+1\right) \ln 2}{\rho^2 \varepsilon^2} \text{ samples} \sim \mathbb{P}\right\}$

Choose $v' \in [v - 2/3\varepsilon, v - 1/3\varepsilon]$ uniformly randomly

Output all $x \in \hat{\mathcal{X}}$ such that $\hat{\mathbb{P}}_S(x) \geq v'$ {empirical distribution}

---

**Theorem D.4.** *Fix $v, \rho \in (0, 1)$ and $\varepsilon \in (0, v), \delta \in (0, \rho/3)$. Then Algorithm D.2 is $\rho$-replicable and returns a list $L$ of elements $x$ such that with probability at least $1 - \delta$:*

    *(a) If $\mathbb{P}(x) \leq v - \varepsilon$, then $x \notin L$.*

    *(b) If $\mathbb{P}(x) \geq v$, then $x \in L$.*

*Moreover, it has sample complexity*

$$\tilde{O}\left(\min\left(|\mathcal{X}|, \frac{1}{(v-\varepsilon)}\right)\frac{1}{\rho^2\varepsilon^2}\log\frac{1}{\delta}\right).$$

*Proof.* First, we wish to capture all $v - \varepsilon$ heavy hitters in $\hat{\mathcal{X}}$. If $\mathcal{X}$ is sufficiently small, this is easy. Otherwise, we fail to observe each $(v - \varepsilon)$ heavy hitter with probability at most $(1 - v + \varepsilon)^{|\hat{\mathcal{X}}|}$. By a union bound over all $(v - \varepsilon)^{-1}$ possible heavy hitters, we fail to capture all $v - \varepsilon$ heavy hitters with probability at most

$$\frac{(1 - v + \varepsilon)^{|\hat{\mathcal{X}}|}}{v - \varepsilon} \leq \frac{1}{v - \varepsilon}\exp\left[-(v - \varepsilon)\cdot\frac{\ln\frac{2}{\delta(v-\varepsilon)}}{v - \varepsilon}\right]$$

$$\leq \frac{\delta}{2}.$$

Moving forward, we condition on this step succeeding.

Next, consider $\hat{p}_1, \hat{p}_2$, the mass estimates over the course of two runs supported on the candidate sets $\hat{\mathcal{X}}_1, \hat{\mathcal{X}}_2$. both follow (possibly different) multinomial distributions of dimension at most $|\hat{\mathcal{X}}| + 1$ with unknown mean parameters $p_1(x)$ for $x \in \hat{\mathcal{X}}_1 \cup \{y_1\}$, $p_2(x)$ for $x \in \hat{\mathcal{X}}_2 \cup \{y_2\}$, respectively, and $|S|$. Here $y_1, y_2$ are dummy elements for observations beyond $\hat{\mathcal{X}}_1, \hat{\mathcal{X}}_2$. Suppose we draw $\frac{\ln 2/\delta + (|\hat{\mathcal{X}}|+1)\ln 2}{2\varepsilon'^2}$ samples over each of two runs to yield estimates $\hat{p}_1, \hat{p}_2$. By Proposition B.1,

$$\sum_{x\in\hat{\mathcal{X}}_1\cup\{y_1\}}|\hat{p}_1(x) - p_1(x)| < 2\varepsilon'$$

$$\sum_{x\in\hat{\mathcal{X}}_2\cup\{y_2\}}|\hat{p}_2(x) - p_2(x)| < 2\varepsilon'$$

each with probability at least $1 - \delta/2$. Moving forward, we condition on this step succeeding.

Now, consider choosing $v' \in [v - 2\varepsilon/3, v - \varepsilon/3]$ uniformly at random.

Correctness: By choosing $2\varepsilon' < \varepsilon/3$, any element $x$ with true mass at least $v$ will have empirical mass strictly more than $v - \varepsilon/3 \geq v'$. Similarly, any element $x$ with true mass at most $v - \varepsilon$ will have empirical mass strictly less than $v - 2\varepsilon/3 \leq v'$. Thus we satisfy the correctness guarantees. Note that we satisfy this guarantee with probability at least $1 - \delta$.

Replicability: $v'$ lands in between some $\hat{p}_1(x), \hat{p}_2(x)$ for $x \in \hat{\mathcal{X}}_1 \cap \hat{\mathcal{X}}_2$ with total probability at most

$$\frac{\sum_{x\in\hat{\mathcal{X}}_1\cap\hat{\mathcal{X}}_2}|\hat{p}_1(x) - \hat{p}_2(x)|}{\varepsilon/3} \leq \frac{\sum_{x\in\hat{\mathcal{X}}_1\cap\hat{\mathcal{X}}_2}|\hat{p}_1(x) - p_1(x)|}{\varepsilon/3} + \frac{\sum_{x\in\hat{\mathcal{X}}_1\cap\hat{\mathcal{X}}_2}|\hat{p}_2(x) - p_2(x)|}{\varepsilon/3}$$

In addition, we wish for $v'$ to avoid landing "below" any $\hat{p}_1(x), \hat{p}_2(x)$ where $x \notin \hat{\mathcal{X}}_1 \cap \hat{\mathcal{X}}_2$. This happens with probability at most

$$\frac{\sum_{x\in\hat{\mathcal{X}}_1\cup\{y_1\}\setminus\hat{\mathcal{X}}_2}|\hat{p}_1(x) - p_1(x)|}{\varepsilon/3} + \frac{\sum_{x\in\hat{\mathcal{X}}_2\cup\{y_2\}\setminus\hat{\mathcal{X}}_1}|\hat{p}_2(x) - p_2(x)|}{\varepsilon/3}.$$

The sum of the two expressions above is at most $\frac{4\varepsilon'}{\varepsilon/3}$. Thus we set

$$\frac{4\varepsilon'}{\varepsilon/3} \leq \frac{\rho}{3}$$

$$\varepsilon' \leq \frac{\rho\varepsilon}{36}.$$

The probability of failing to be replicable is then at most $1 - \rho/3 - 2\delta \geq 1 - \rho$.

Sample Complexity: Recall that we set

$$\varepsilon' = \min\left(\frac{\rho\varepsilon}{36}, \frac{\varepsilon}{6}\right) = \frac{\rho\varepsilon}{36}.$$

If $|\mathcal{X}|$ is sufficiently small, the sample complexity becomes

$$\frac{\ln \frac{2}{\delta} + (|\mathcal{X}| + 1) \ln 2}{2\varepsilon'^2} = \frac{648 \ln \frac{2}{\delta} + 648 (|\mathcal{X}| + 1) \ln 2}{\rho^2 \varepsilon^2}$$

Otherwise, the sample complexity is

$$\frac{\ln \frac{2}{\delta(v-\varepsilon)}}{v - \varepsilon} + \frac{648 \ln \frac{2}{\delta}}{\rho^2 \varepsilon^2} + \frac{648 \left( \ln \frac{2}{\delta(v-\varepsilon)} + 1 \right) \ln 2}{\rho^2 \varepsilon^2 (v - \varepsilon)}.$$

$\square$

## D.3 Replicable Coreset

We now prove that Algorithm 4.1 replicably yields an $\varepsilon$-coreset. In Appendix D.3.1, we state Algorithm D.3, an equivalent description of Algorithm 4.1. Then, we show that it is indeed replicable in Appendix D.3.2. Following this, we prove that the cost of any set of centers $F \subseteq \mathcal{B}_d$ is preserved up to an $\varepsilon$-multiplicative error in Appendix D.3.3. In Appendix D.3.4, we show that the outputted coreset is of modest size. Last but not least, we summarize our findings in Appendix D.3.5.

### D.3.1 Pseudocode

Before presenting the analysis, we state an equivalent algorithm to Algorithm 4.1 which uses precise notation that is more suitable for analysis.

Starting with the grid that consists of a single cell $[-1/2, 1/2]^d$ at layer 0, we recursively subdivide each cell into $2^d$ smaller cells of length $2^{-i}$ at layer $i$. Each subdivided cell is a *child* of the larger cell, the containing cell is the *parent* of a child cell, and cells sharing the same parent are *siblings*. Let $\mathcal{G}_i$ denote the union of cells at the $i$-th level. We write $\mathbb{P}_i$ to denote the discretized distribution to $\mathcal{G}_i$. In other words, $\mathbb{P}_i = \mathbb{P}\big|_{\sigma(\mathcal{G}_i)}$ is the restriction of $\mathbb{P}$ to the smallest $\sigma$-algebra containing $\mathcal{G}_i$. We say a cell on the $i$-th level is *heavy* if the replicable heavy hitters algorithm (cf. Algorithm D.2) returns it with input distribution $\mathbb{P}_i$, heavy hitter threshold $v = \frac{\gamma \cdot \Lambda}{2^{-pi}}$, and error $v/2$ for some parameter $\gamma$ to be determined later. A cell that is not heavy is *light*. The recursion terminates either when there are no more heavy cells on the previous layer, or the current grid length is a sufficiently small fraction of OPT. In the second case, we say the last layer consists of *special* cells.

If no child cell of a heavy parent is heavy, we mark the parent. Otherwise, we recurse on its heavy children so that one of its descendants will eventually be marked. The recursion must stop, either when there are no more heavy children or when the side length is no more than a fraction of $\Lambda$. Note that the light cells with heavy parents and special cells partition $[-1/2, 1/2]^d$.

Then, we build $R$ in reverse as follows. For a marked heavy cell $Z$, we set $R(x)$ to be an arbitrary (but replicable) point in $Z$, e.g. its center, for all $x \in Z$. This covers all light cells with no heavy siblings as well as special cells. Otherwise, a cell $Z$ is light with heavy siblings, and its parent is heavy but unmarked. Thus, the parent has some marked heavy descendent $Z'$, and we choose $R(x) = R(x')$ for all $x \in Z$ and $x' \in Z'$.

### D.3.2 Replicability

**Proposition D.5.** *Algorithm D.3 terminates after at most*

$$t := \left\lceil \frac{1}{p} \log \frac{10\beta\Delta^p}{\varepsilon \cdot \text{OPT}} + 1 \right\rceil$$

*layers.*

---

**Algorithm D.3** Replicable Coreset; Algorithm 4.1 Equivalent

---

1: **rCoreset**(distribution $\mathbb{P}$, accuracy $\varepsilon$, exponent $p$, replicability $\rho$, confidence $\delta$):
2: Init $\mathcal{H}[i] \leftarrow \varnothing$ for all $i$
3: Init $\mathcal{L}[i] \leftarrow \varnothing$ for all $i$
4: Init $\mathcal{S}[i] \leftarrow \varnothing$ for all $i$
5:
6: $\mathcal{H}(0) \leftarrow \left\{ [-1/2, 1/2]^d \right\}$
7: **for** $i \leftarrow 1; H[i-1] \neq \varnothing; i \leftarrow i+1$ **do**
8:    **if** $(2^{-i+1}\Delta)^p \leq \varepsilon\Lambda/5$ **then**
9:       $\mathcal{S}[i] \leftarrow \text{children}(\mathcal{H}[i-1])$
10:    **else**
11:       $\mathcal{H}[i] \leftarrow \text{children}(\mathcal{H}[i-1]) \cap \text{rHeavyHitters}\left(\mathbb{P}_i, v = \frac{\gamma \cdot \Lambda}{2^{-pi}}, v/2, \rho/t, \delta/t\right)$ {$t$ is an upper

      bound on the number of layers.}
12:       $\mathcal{L}[i] \leftarrow \text{children}(\mathcal{H}[i-1]) \setminus \mathcal{H}[i]$
13:    **end if**
14: **end for**
15:
16: **for** $j \leftarrow i-1; j \geq 0; j \leftarrow j-1$ **do**
17:    **for** $Z \in \mathcal{H}[j]$ **do**
18:       **if** $\text{children}(Z) \cap \mathcal{H}[j+1] = \varnothing$ **then**
19:          $R'(Z) \leftarrow \text{rChoice}(Z)$
20:       **else**
21:          $R'(Z) \leftarrow R'(\text{rChoice}(\text{children}(Z) \cap \mathcal{H}[j+1]))$
22:       **end if**
23:    **end for**
24:    **for** $Z \in \mathcal{L}[j+1]$ **do**
25:       $R(Z) \leftarrow R'(\text{parent}(Z))$
26:    **end for**
27:    **for** $Z \in \mathcal{S}[j+1]$ **do**
28:       $R(Z) \leftarrow R'(\text{parent}(Z))$
29:    **end for**
30: **end for**
31:
32: Output $R$

---

*Proof.* We have

$$(2^{-i+1}\Delta)^p \leq \frac{\varepsilon\Lambda}{5}$$

$$\iff 2^{p(-i+1)} \leq \frac{\varepsilon\Lambda}{5\Delta^p}$$

$$\iff p(-i+1) \leq \log \frac{\varepsilon\Lambda}{5\Delta^p}$$

$$\iff i \geq \frac{1}{p} \log \frac{5\Delta^p}{\varepsilon \cdot \Lambda} + 1.$$

But $1/\Lambda \leq \beta(1+\varepsilon)/\text{OPT}$, concluding the proof. $\square$

**Corollary D.6.** *Algorithm D.3 is $\rho$-replicable and succeeds with probability at least $1 - \delta$. Moreover, it has sample complexity*

$$\tilde{O}\left(\frac{t^2}{\rho^2\gamma^3 \cdot \text{OPT}^3} \log \frac{1}{\delta}\right).$$

*Proof.* The only non-trivial random components of Algorithm D.3 is the heavy hitter estimation. By Theorem D.4, the heavy hitters subroutine call on the $i$-th layer is $\rho/t$-replicable and succeeds with

probability at least $1 - \delta/t$. Also, it has sample complexity

$$\tilde{O}\left(\frac{t^2 \cdot 2^{3(-pi+1)}}{\rho^2 \gamma^3 \cdot \Lambda^3} \log \frac{1}{\delta}\right).$$

Now,

$$\sum_{i=0}^{t} 2^{-3pi} \le \sum_{i=0}^{\infty} 2^{-3pi}$$
$$= O(1).$$

Thus all in all, Algorithm D.3 is $\rho$-replicable and succeeds with probability at least $1 - \delta$. It also has sample complexity

$$\tilde{O}\left(\frac{t^2}{\rho^2 \gamma^3 \cdot \Lambda^3} \log \frac{1}{\delta}\right).$$

Since $1/\Lambda \le \beta(1+\varepsilon)/\text{OPT}$, we conclude the proof. $\qquad\square$

### D.3.3 Expected Shift

**Proposition D.7.** *For any $x \in Z \in \mathcal{L}[i] \cup \mathcal{S}[i]$ outputted by Algorithm D.3,*

$$\kappa(x, R(x)) \le (2^{-i+1}\Delta).$$

*Proof.* $x$ is represented by some point within its parent. $\qquad\square$

**Proposition D.8.** *The output $\mathcal{S}[t]$ of Algorithm D.3 satisfies*

$$\int_{\mathcal{S}[t]} \kappa(x, R(x))^p d\mathbb{P}(x) \le \frac{\varepsilon\Lambda}{5}.$$

*Proof.* By computation,

$$\begin{aligned}
\int_{\mathcal{S}[t]} \kappa(x, R(x))^p d\mathbb{P}(x) &\le \int_{\mathcal{S}[t]} (2^{-i+1}\Delta)^p d\mathbb{P}(x) && \text{Proposition D.7} \\
&\le \int_{\mathcal{S}[t]} \frac{\varepsilon\Lambda}{5} d\mathbb{P}(x) && \text{definition of } t \\
&\le \int_{\mathcal{X}} \frac{\varepsilon\Lambda}{5} d\mathbb{P}(x) \\
&= \frac{\varepsilon\Lambda}{5}. && \square
\end{aligned}$$

Let $F$ be an arbitrary set of $k$ centers. Partition $\mathcal{L}[i]$ into

$$\mathcal{L}_{\text{near}}[i] := \left\{ Z \in \mathcal{L}[i] : \min_{x \in Z} \kappa(x, F)^p \le \frac{5}{\varepsilon}(2^{-i+1}\Delta)^p \right\}$$

$$\mathcal{L}_{\text{far}}[i] := \left\{ Z \in \mathcal{L}[i] : \min_{x \in Z} \kappa(x, F)^p > \frac{5}{\varepsilon}(2^{-i+1}\Delta)^p \right\}.$$

**Proposition D.9.** *The output $\mathcal{L}_{\text{far}}$ of Algorithm D.3 satisfies*

$$\sum_{i=0}^{t} \int_{\mathcal{L}_{\text{far}}[i]} \kappa(x, R(x))^p d\mathbb{P}(x) \le \frac{\varepsilon}{5} \text{cost}(F).$$

*Proof.* Any $x \in Z \in \mathcal{L}_{\text{far}}[i]$ contributes a cost of at least $\frac{5}{\varepsilon}(2^{-i+1}\Delta)^p$. Hence

$$\sum_{i=0}^{t} \int_{\mathcal{L}_{\text{far}}[i]} \kappa(x, R(x))^p d\mathbb{P}(x) \leq \sum_{i=0}^{t} \int_{\mathcal{L}_{\text{far}}[i]} (2^{-i+1}\Delta)^p d\mathbb{P}(x)$$

$$= \frac{\varepsilon}{5} \sum_{i=0}^{t} \int_{\mathcal{L}_{\text{far}}[i]} \frac{5}{\varepsilon}(2^{-i+1}\Delta)^p d\mathbb{P}(x)$$

$$\leq \frac{\varepsilon}{5} \sum_{i=0}^{t} \int_{\mathcal{L}_{\text{far}}[i]} \kappa(x, F)^p d\mathbb{P}(x)$$

$$\leq \frac{\varepsilon}{5} \operatorname{cost}(F). \qquad \square$$

**Proposition D.10.** *The output $\mathcal{L}_{\text{near}}$ of Algorithm D.3 satisfies $|\mathcal{L}_{\text{near}}[i]| \leq kM$ where*

$$M := \left[ \frac{2^5}{\varepsilon} \Delta \right]^d$$

*Proof.* The furthest point $x$ in a cell in $\mathcal{L}_{\text{near}}[i]$ can have a distance of at most

$$2^{-i}\Delta + \sqrt[p]{\frac{5}{\varepsilon}}(2^{-i+1}\Delta) \leq \left(1 + \frac{10}{\varepsilon}\right)(2^{-i}\Delta)$$

to the nearest center. Thus the points belonging to a particular center live in an $\ell_\infty$ ball of that radius. Not including the cell containing the center, we can walk past at most

$$\frac{1}{2^{-i}} \cdot \left(1 + \frac{10}{\varepsilon}\right)(2^{-i}\Delta) = \left(1 + \frac{10}{\varepsilon}\right)\Delta$$

cells if we walk in the directions of the canonical basis of $\mathbb{R}^d$. It follows that there can be at most

$$\left[1 + 2\left(1 + \frac{10}{\varepsilon}\right)\Delta\right]^d \leq \left[\frac{32}{\varepsilon}\Delta\right]^d$$

cells in $\mathcal{L}_{\text{near}}[i]$ close to each center of $F$, and thus there are at most $kM$ cells in total. $\qquad \square$

**Proposition D.11.** *By choosing*

$$\gamma := \frac{\varepsilon}{5tkM(2\Delta)^p},$$

*The output $\mathcal{L}_{\text{near}}$ of Algorithm D.3 satisfies*

$$\sum_{i=0}^{t} \int_{\mathcal{L}_{\text{near}}[i]} \kappa(x, R(x))^p \leq \frac{\varepsilon\Lambda}{5}.$$

*Proof.* Each light cell has measure at most $\frac{\gamma \cdot \Lambda}{2^{-pi}}$. By computation,

$$\sum_{i=0}^{t} \int_{\mathcal{L}_{\text{near}}[i]} \kappa(x, R(x))^p d\mathbb{P}(x) \leq \sum_{i=1}^{t} \int_{\mathcal{L}_{\text{near}}[i]} (2^{-i+1}\Delta)^p d\mathbb{P}(x)$$

$$\leq \sum_{i=1}^{t} kM \cdot \frac{\gamma \cdot \Lambda}{2^{-pi}} 2^{-pi}(2\Delta)^p$$

$$= t \cdot kM \cdot \gamma \cdot \Lambda \cdot (2\Delta)^p$$

$$\leq \frac{\varepsilon\Lambda}{5}. \qquad \square$$

**Corollary D.12.** *Set*

$$\gamma := \frac{\varepsilon}{5tkM(2\Delta)^p}.$$

*Then for any set $F$ of $k$-centers, the output $R$ of Algorithm D.3 satisfies*

$$\int_{\mathcal{X}} \kappa(x, R(x))^p d\mathbb{P}(x) \leq \varepsilon \operatorname{cost}(F).$$

*Proof.* $\mathcal{X}$ is contained in the union of all $\mathcal{L}[i]$'s and $\mathcal{S}[t]$. Hence

$$
\begin{aligned}
\int_{\mathcal{X}} \kappa(x, R(x))^p d\mathbb{P}(x) &\leq \frac{2\varepsilon\Lambda}{5} + \frac{\varepsilon}{5}\operatorname{cost}(F) \\
&\leq \frac{2\varepsilon(1+\varepsilon)\operatorname{OPT}}{5} + \frac{\varepsilon}{5}\operatorname{cost}(F) \\
&\leq \frac{4\varepsilon}{5}\operatorname{OPT} + \frac{\varepsilon}{5}\operatorname{cost}(F) \\
&\leq \varepsilon\operatorname{cost}(F). \qquad \square
\end{aligned}
$$

### D.3.4 Coreset Size

To determine the size of our coreset, we bound the number of marked heavy cells, as all our representative points are in marked heavy cells.

Fix $F_{\mathrm{OPT}}$ to be a set of optimal centers for the $(k, p)$-clustering problem. Write $\mathcal{M}[i]$ to denote the set of marked heavy cells on the $i$-th layer and partition $\mathcal{M}[i]$ as

$$
\mathcal{M}_{\mathrm{close}}[i] := \left\{ Z \in \mathcal{M}[i] : \min_{x \in Z} \kappa(x, F) \leq 2^{-pi+1} \right\}
$$

$$
\mathcal{M}_{\mathrm{dist}}[i] := \left\{ Z \in \mathcal{M}[i] : \min_{x \in Z} \kappa(x, F) > 2^{-pi+1} \right\}.
$$

**Proposition D.13.** *The marked heavy cells $\mathcal{M}_{\mathrm{dist}}$ outputted by Algorithm D.3 satisfy*

$$
\left| \bigcup_{0 \leq i \leq t} \mathcal{M}_{\mathrm{dist}}[i] \right| \leq \frac{2\beta}{\gamma}.
$$

*Proof.* Each cell in $\mathcal{M}_{\mathrm{dist}}[i]$ has mass at least $\frac{\gamma \cdot \Lambda}{2^{-pi+1}}$ and each $x \in Z \in \mathcal{M}_{\mathrm{dist}}[i]$ is of distance at least $2^{-pi+1}$ to its closest center. Thus each cell contributes at least

$$
\frac{\gamma \cdot \Lambda}{2^{-pi+1}} \cdot 2^{-pi+1} = \gamma \cdot \Lambda
$$

to the objective. If follows that there are at most

$$
\frac{\operatorname{OPT}}{\gamma\Lambda} \leq \frac{1}{\gamma}\beta(1+\varepsilon) \leq \frac{2\beta}{\gamma}.
$$

such cells. $\qquad \square$

**Proposition D.14.** *The marked heavy cells $\mathcal{M}_{\mathrm{close}}$ outputted by Algorithm D.3 satisfy*

$$
|\mathcal{M}_{\mathrm{close}}[i]| \leq k(7\Delta)^d.
$$

*Proof.* The furthest point $x$ in a cell $\mathcal{M}_{\mathrm{close}}[i]$ to its nearest center is

$$
2^{-pi+1} + 2^{-i}\Delta.
$$

In other words, not including the cell containing the center, we can walk past at most

$$
2^{i(1-p)+1} + \Delta \leq 2 + \Delta
$$

cells by walking along the direction of the canonical basis of $\mathbb{R}^d$. Thus there are at most

$$
k\left[1 + 2\left(2 + \Delta\right)\right]^d \leq k\left[7\Delta\right]^d
$$

such cells on the $i$-th layer. $\qquad \square$

**Proposition D.15.** *The image of the map $R$ outputted by Algorithm D.3 satisfies*

$$
N := |R(\mathcal{X})| \leq \frac{2\beta}{\gamma} + tk(7\Delta)^d.
$$

*In particular, for*

$$\gamma := \frac{\varepsilon}{5tkM(2\Delta)^p},$$

*we have*

$$N := |R(\mathcal{X})| \leq \frac{3\beta}{\gamma} = 3\beta \cdot \frac{5tkM(2\Delta)^p}{\varepsilon}.$$

*Proof.* The image is contained in the marked heavy cells. □

### D.3.5  $k$-Medians & $k$-Means

We now derive results for the case of $p = 1$.

**Theorem D.16.** *Algorithm D.3 is $\rho$-replicable and outputs an $\varepsilon$-coreset of size*

$$O\left(\frac{k2^{5d}\Delta^{d+1}}{\varepsilon^{d+1}} \log \frac{\Delta}{\varepsilon \cdot \text{OPT}}\right)$$

*for statistical $k$-medians with probability at least $1 - \delta$. Moreover, it has sample complexity*

$$\tilde{O}\left(\frac{k^3 2^{15d}\Delta^{3d+3}}{\rho^2 \varepsilon^{3d+3} \cdot \text{OPT}^3} \log \frac{1}{\delta}\right).$$

*Proof (Theorem D.16).* By Corollary D.12, Algorithm D.3 outputs an $\varepsilon$-coreset. By Corollary D.6, Algorithm D.3 is $\rho$-replicable with sample complexity

$$\tilde{O}\left(\frac{t^2}{\rho^2 \gamma^3 \cdot \text{OPT}^3} \log \frac{1}{\delta}\right).$$

We have

$$\frac{3\beta}{\gamma} = O\left(\frac{k\left[2^5\Delta/\varepsilon\right]^d (2\Delta)}{\varepsilon} \cdot \log \frac{\Delta}{\varepsilon \cdot \text{OPT}}\right) = O\left(\frac{k2^{5d}\Delta^{d+1}}{\varepsilon^{d+1}} \log \frac{\Delta}{\varepsilon \cdot \text{OPT}}\right)$$

By Proposition D.15, the coreset has size at most this value.

Substituting the value of $\gamma$ above and remarking that $t$ is a polylogarithmic factor of the other terms, we thus conclude that the final sample complexity is

$$\tilde{O}\left(\frac{k^3 2^{15d}\Delta^{3d+3}}{\varepsilon^{3d+3}} \cdot \frac{1}{\rho^2 \cdot \text{OPT}^3} \log \frac{1}{\delta}\right) = \tilde{O}\left(\frac{k^3 2^{15d}\Delta^{3d+3}}{\rho^2 \varepsilon^{3d+3} \cdot \text{OPT}^3} \frac{1}{\delta}\right). \qquad \square$$

Now consider the case of $p = 2$. We can perform our analysis with the help of a standard inequality.

**Proposition D.17.** *For any set of centers $F$,*

$$\mathbb{E}_x \kappa(R(x), F)^2 \leq \mathbb{E}_x \kappa(x, F)^2 + 2\sqrt{\mathbb{E}_x\left[\kappa(x, F)^2\right] \cdot \mathbb{E}_x\left[\kappa(x, R(x))^2\right]} + \mathbb{E}_x \kappa(x, R(x))^2$$

$$\mathbb{E}_x \kappa(x, F)^2 \leq \mathbb{E}_x \kappa(R(x), F)^2 + 2\sqrt{\mathbb{E}_x\left[\kappa(R(x), F)^2\right] \cdot \mathbb{E}_x\left[\kappa(x, R(x))^2\right]} + \mathbb{E}_x \kappa(x, R(x))^2.$$

*Thus if $\mathbb{E}_x\kappa(x, R(x))^2 \leq \frac{\varepsilon^2}{2^6}\mathbb{E}_x\kappa(x, F)^2$,*

$$\mathbb{E}_x \kappa(R(x), F)^2 \leq \mathbb{E}_x \kappa(x, F)^2 + 2 \cdot \frac{1}{2^3}\varepsilon\mathbb{E}_x\left[\kappa(x, F)^2\right] + \frac{1}{2^6}\varepsilon^2\mathbb{E}_x\left[\kappa(x, F)^2\right]$$

$$\leq \left(1 + \frac{\varepsilon}{4}\right)\mathbb{E}_x\kappa(x, F)^2$$

$$\mathbb{E}_x \kappa(x, F)^2 \leq \mathbb{E}_x \kappa(R(x), F)^2 + 2\sqrt{(1 + \varepsilon)\mathbb{E}_x\left[\kappa(x, F)^2\right] \cdot \frac{1}{2^6}\varepsilon^2\mathbb{E}_x\left[\kappa(x, F)^2\right]} + \frac{1}{2^6}\varepsilon^2\mathbb{E}_x\kappa(x, F)^2$$

$$\leq \mathbb{E}_x\kappa(R(x), F)^2 + \frac{1}{2}\varepsilon\mathbb{E}_x\left[\kappa(x, F)^2\right] + \frac{1}{2^6}\varepsilon^2\mathbb{E}_x\kappa(x, F)^2$$

$$\leq \mathbb{E}_x\kappa(R(x), F)^2 + \varepsilon\mathbb{E}_x\kappa(x, F)^2.$$

*Proof.* By Hölder's inequality,

$$
\begin{aligned}
\mathbb{E}_x \kappa(R(x), F)^2 &\le \mathbb{E}_x \left[ \kappa(x, F) + \kappa(x, R(x)) \right]^2 \\
&= \mathbb{E}_x \kappa(x, F)^2 + 2\mathbb{E}_x[\kappa(x, F)\kappa(x, R(x))] + \mathbb{E}_x \kappa(x, R(x))^2 \\
&\le \mathbb{E}_x \kappa(x, F)^2 + 2\sqrt{\mathbb{E}_x \left[\kappa(x, F)^2\right] \mathbb{E}_x \left[\kappa(x, R(x))^2\right]} + \mathbb{E}_x \kappa(x, R(x))^2.
\end{aligned}
$$

The other case is identical. $\qquad\square$

**Theorem D.18.** *Algorithm D.3 is $\rho$-replicable and outputs an $\varepsilon$-coreset for statistical $k$-means of size*

$$
O\left( \frac{k 2^{11d} \Delta^{d+2}}{\varepsilon^{2d+2}} \log \frac{\Delta^2}{\varepsilon \operatorname{OPT}} \right)
$$

*with probability at least $1 - \delta$. Moreover, it has sample complexity*

$$
\tilde{O}\left( \frac{k^3 2^{33d} \Delta^{3d+6}}{\rho^2 \varepsilon^{6d+6} \cdot \operatorname{OPT}^3} \log \frac{1}{\delta} \right).
$$

*Proof (Theorem D.18).* Similar to $k$-medians, Algorithm D.3 outputs an $\varepsilon$-coreset with probability at least $1 - \delta$, is $\rho$-replicable, and has sample complexity

$$
\tilde{O}\left( \frac{t^2}{\rho^2 \gamma^3 \cdot \operatorname{OPT}^3} \log \frac{1}{\delta} \right).
$$

However, we need to take $\varepsilon' := \varepsilon^2/2^6$.

We have

$$
\frac{3}{\gamma} = O\left( \frac{tk \left[ 2^{5+6}\Delta/\varepsilon^2 \right]^d (2\Delta)^2}{\varepsilon^2} \right) = O\left( \frac{k 2^{11d} \Delta^{d+2}}{\varepsilon^{2d+2}} \log \frac{\Delta^2}{\varepsilon \operatorname{OPT}} \right).
$$

By Proposition D.15, the coreset has size at most this value.

$t$ is a polylogarithmic factor of the rest of the parameters, thus we thus conclude that the final sample complexity is

$$
\tilde{O}\left( \frac{k^3 2^{33d} \Delta^{3d+6}}{\varepsilon^{6d+6}} \cdot \frac{1}{\rho^2 \cdot \operatorname{OPT}^3} \right) = \tilde{O}\left( \frac{k^3 2^{33d} \Delta^{3d+6}}{\rho^2 \varepsilon^{6d+6} \cdot \operatorname{OPT}^3} \log \frac{1}{\delta} \right). \qquad\square
$$

It is not clear how to compare our guarantees with that of Frahling and Sohler [2005] as we attempt to minimize the number of samples required from a distribution assuming only sample access. On the other hand, Frahling and Sohler [2005] attempted to minimize the running time of their algorithm assuming access to the entire distribution.

### D.4 Replicable Multinomial Parameter Estimation

Now that we have reduced the problem to a finite distribution (coreset), we can think of it as a weighted instance of $(k, p)$-clustering where the weights are unknown but can be estimated using data.

**Proposition D.19.** *Enumerate the coreset $R(\mathcal{X}) = r^{(1)}, \dots, r^{(N)}$ and let $w^{(i)}$ be the probability mass at $r^{(i)}$. If we have estimates $\hat{w}^{(i)}$ satisfying*

$$
|\hat{w}^{(i)} - w^{(i)}| < \frac{\varepsilon}{N},
$$

*then*

$$
\left| \sum_{i=1}^{N} w^{(i)} \kappa(r^{(i)}, F)^p - \sum_{i=1}^{N} \hat{w}^{(i)} \kappa(r^{(i)}, F)^p \right| \le \varepsilon.
$$

*for all centers $F$.*

*Proof.* Indeed,

$$
\left| \sum_{i=1}^{N} w^{(i)} \kappa(r^{(i)}, F)^p - \sum_{i=1}^{N} \hat{w}^{(i)} \kappa(r^{(i)}, F)^p \right| \leq \sum_{i=1}^{N} |w^{(i)} - \hat{w}^{(i)}| \kappa(r^{(i)}, F)^p
$$

$$
\leq \sum_{i=1}^{N} |w^{(i)} - \hat{w}^{(i)}| \cdot 1
$$

$$
\leq \varepsilon. \qquad \square
$$

Let us formulate this as a replicable multinomial parameter estimation problem. Consider a multinomial distribution $Z = (Z^{(1)}, \ldots, Z^{(N)})$ of dimension $N$ with parameters $p^{(1)}, \ldots, p^{(N)}$ and $n$. Note that this can be cast as a more general statistical query problem which we illustrate below. We allow simultaneous estimation of more general statistical queries $g_1, \ldots, g_N : (\mathcal{X}^n, \mathbb{P}) \to \mathbb{R}$ compared to Impagliazzo et al. [2022], assuming they can be estimated with high accuracy and confidence from data, say each query $g_j(x_1, \ldots, x_n)$ concentrates about its true value $G_j \in \mathbb{R}$ with high probability.

---

**Algorithm D.4** Replicable Rounding

---

1: **rRounding**(statistical queries $g_1, \ldots g_N$, distribution $\mathbb{P}$, number of samples $n$, error $\varepsilon$, replicability $\rho$, confidence $\delta$):
2: Sample $x_1, \ldots, x_n \sim \mathbb{P}$
3: $\alpha \leftarrow \frac{2\varepsilon}{\rho + 1 - 2\delta}$
4: **for** $j \leftarrow 1, \ldots, N$: **do**
5:     $\alpha^{(j)} \leftarrow$ uniform random sample $[0, \alpha]$
6:     Split $\mathbb{R}$ into intervals $I^{(j)} = \{[\alpha^{(j)} + z\alpha, \alpha^{(j)} + (z+1)\alpha) : z \in \mathbb{Z}\}$
7:     Output the midpoint $\hat{G}_j$ of the interval within $I^{(i)}$ that $g_j(x_1, \ldots, x_n)$ falls into
8: **end for**

---

**Theorem D.20** (Replicable Rounding). *Suppose we have a finite class of statistical queries $g_1, \ldots, g_N$ and sampling $n$ independent points from $\mathbb{P}$ ensures that*

$$
\sum_{j=1}^{N} |g_j(x_1, \ldots, x_n) - G_j| \leq \varepsilon' := \frac{\varepsilon(\rho - 2\delta)}{\rho + 1 - 2\delta}
$$

*with probability at least $1 - \delta$.*

*Then Algorithm D.4 is $\rho$-replicable and outputs estimates $\hat{G}_j$ such that*

$$
|\hat{G}_j - G_j| \leq \varepsilon
$$

*with probability at least $1 - \delta$ for every $j \in [N]$. Moreover, it requires at most $n$ samples.*

*Proof.* Outputting the midpoint of the interval can offset each $g_j(x_1, \ldots, x_n)$ by at most $\frac{\alpha}{2}$. Hence

$$
|\hat{G}_j - G_j| \leq \frac{\varepsilon(\rho - 2\delta)}{\rho + 1 - 2\delta} + \frac{\varepsilon}{\rho + 1 - 2\delta} = \varepsilon.
$$

Consider two executions of our algorithm yielding estimates $\hat{G}^{(1)}, \hat{G}^{(2)}$. The probability that any of the estimates fail to satisfy $\ell_1$-tolerance $\varepsilon'$ is at most $2\delta$. The two executions output different estimates only if some random offset "splits" some $\hat{G}_j^{(1)}, \hat{G}_j^{(2)}$. Conditioning on the success to satisfy $\ell_1$-tolerance, this occurs with probability at most

$$
\sum_{j=1}^{N} \frac{|\hat{G}_j^{(1)} - \hat{G}_j^{(2)}|}{\alpha} \leq \sum_{j=1}^{N} \frac{|\hat{G}_j^{(1)} - G_j| + |\hat{G}_j^{(2)} - G_j|}{\alpha}
$$

$$
\leq \frac{2\varepsilon'}{\alpha}
$$

$$
\leq \rho - 2\delta.
$$

Accounting for the probability of failure to satisfy the $\ell_1$-tolerance of $2\delta$, our algorithm is $\rho$-replicable.

$\square$

We note that Theorem D.20 can be thought of as a generalization of the SQ oracle (cf. Theorem D.2) from Impagliazzo et al. [2022]. Indeed, simply using Theorem D.2 yields an extra factor of $N$ in the sample complexity as we must take a union bound.

**Corollary D.21.** *Let $\varepsilon, \rho \in (0, 1)$ and $\delta \in (0, \rho/3)$. There is a $\rho$-replicable algorithm that outputs parameter estimates $\bar{p}$ for a multinomial distribution of dimension $N$ such that*

*(a) $|\bar{p}^{(i)} - p^{(i)}| \leq \varepsilon$ for every $i \in [N]$ with probability at least $1 - \delta$.*

*(b) $\bar{p}^{(i)} \geq 0$ for all $i \in [N]$.*

*(c) $\sum_i \bar{p}^{(i)} = 1$.*

*Moreover, the algorithm has sample complexity*

$$O\left(\frac{\ln 1/\delta + N}{\varepsilon^2(\rho - \delta)^2}\right) = \tilde{O}\left(\frac{N}{\varepsilon^2\rho^2}\log\frac{1}{\delta}\right).$$

*Proof.* Define

$$\varepsilon' := \frac{\varepsilon(\rho - 2\delta)}{\rho + 1 - 2\delta}.$$

By Proposition B.1, sampling

$$n = \frac{2\ln 1/\delta + 2N\ln 2}{\varepsilon'^2}$$
$$\leq \frac{8\ln 1/\delta + 8N\ln 2}{\varepsilon^2(\rho - 2\delta)^2}.$$

points implies that $\sum_{i=1}^{N}|\hat{p}^{(i)} - p^{(i)}| < \varepsilon'$ with probability at least $1 - \delta$.

Thus running Algorithm D.4 with functions

$$g_j(x_1, \ldots, x_N) := \frac{1}{N}\sum_{i=1}^{N}\mathbb{1}\{x_i = j\}$$

yields $\bar{p}^{(i)}$'s such that $|\bar{p}^{(i)} - p^{(i)}| \leq \varepsilon$ for each $i \in [N]$. If there are any negative $\bar{p}^{(i)}$'s, we can only improve the approximation by setting them to 0. If the sum of estimates is not equal to 1, we can normalize by taking

$$\bar{p}^{(i)} \leftarrow \bar{p}^{(i)} - \frac{1}{N}\left[\sum_i \bar{p}^{(i)} - 1\right].$$

This introduces an additional error of $\varepsilon/N$ for each $\hat{p}^{(i)}$ and a maximum error of $2\varepsilon$. Choosing $\varepsilon_1 := \varepsilon/2$ concludes the proof. $\square$

## D.5 Replicable OPT Estimation for $(k, p)$-Clustering

Our coreset algorithm assumes the knowledge of a constant ratio estimation of OPT. If this is not provided to us, we show in this section that it is possible to replicably compute such an estimate using a two-step approach. First, we are able to produce replicable estimates with additive error $\varepsilon$ simply by approximately solving the sample $(k, p)$-clustering problem on a sufficiently large sample thanks to uniform convergence (cf. Theorem C.9). Then, we repeat this process with $\varepsilon \leftarrow \varepsilon/2$ until $\varepsilon$ is but a small fraction of the outputted estimate.

**Theorem D.22** (OPT Estimation with Additive Error). *Let $\beta \geq 1$ be an absolute constant and $\varepsilon, \rho \in (0, 1), \delta \in (0, \rho/3)$. Suppose we are provided with an algorithm $\mathcal{A}$ that, given an instance of sample $(k, p)$-clustering, outputs an estimate $\Xi$ of the value $\widehat{\text{OPT}}$ such that*

$$\Xi \in \left[\frac{1}{\beta}\widehat{\text{OPT}}, \widehat{\text{OPT}}\right].$$

*Then there is a $\rho$-replicable algorithm which produces an estimate $\Lambda$ of* OPT *such that*

$$\Lambda \in \left[ \frac{\text{OPT}}{\beta} - \varepsilon, \text{OPT} + \varepsilon \right]$$

*with probability at least $1 - \delta$. Moreover, the algorithm has sample complexity*

$$\tilde{O}\left( \frac{k^2 d^2}{\varepsilon^6 \rho^6} \log \frac{p}{\delta} \right).$$

We note that the algorithm $\mathcal{A}$ can be taken to be any $\beta$-approximation algorithm with some postprocessing, where we divide the value of the cost by $\beta$. Alternatively, we can take $\mathcal{A}$ to be some convex relaxation of our problem with constant integral gap.

*Proof.* Let $X_1, \ldots, X_N \sim \mathbb{P}$ be i.i.d. random variables. The output $\Xi = \mathcal{A}(X_1, \ldots, X_N) \in [0, 1]$ is thus a bounded random variable. Suppose we repeat this experiment $n$ times to produce estimates $\Xi_1, \ldots, \Xi_n$. By an Hoeffding bound,

$$\mathbb{P}\left\{ \frac{1}{n} \sum_{i=1}^{n} (\Xi_i - \mathbb{E}[\Xi]) > \varepsilon \right\} \leq 2 \exp\left( -2n\varepsilon^2 \right).$$

Thus with

$$n \geq \frac{1}{2\varepsilon^2} \ln \frac{2}{\delta}$$

trials, the average estimate $\bar{\Xi} := \frac{1}{n} \sum_{i=1}^{n} \Xi_i$ satisfies $\left| \bar{\Xi} - \mathbb{E}[\Xi] \right| < \varepsilon$ with probability at least $1 - \delta$. Now, by Theorem C.9, choosing

$$N = O\left( \frac{k^2 d^2}{\varepsilon^4} \log \frac{np}{\delta} \right)$$

i.i.d. samples from $\mathbb{P}$ ensures that $\left| \widehat{\text{OPT}} - \text{OPT} \right| \leq \varepsilon$ for all $n$ trial with probability at least $1 - \delta$. Thus conditioning on success, we have

$$\Xi_i \in \left[ \frac{\text{OPT}}{\beta} - \varepsilon, \text{OPT} + \varepsilon \right]$$

for every trial $i \in [n]$. But then the average $\bar{\Xi}$ also falls into this interval as well.

We have shown that there is an algorithm that outputs an estimate

$$\bar{\Xi} \in [\mathbb{E}[\Xi] - \varepsilon, \mathbb{E}[\Xi] + \varepsilon] \cap \left[ \frac{\text{OPT}}{\beta} - \varepsilon, \text{OPT} + \varepsilon \right]$$

with probability at least $1 - \delta$. Moreover, it has sample complexity

$$nN = O\left( \frac{1}{\varepsilon^2} \log \frac{1}{\delta} \cdot \frac{k^2 d^2}{\varepsilon^4} \log \left( \frac{p}{\varepsilon^2 \delta} \log \frac{1}{\delta} \right) \right)$$

$$= O\left( \frac{k^2 d^2}{\varepsilon^6} \log^2 \frac{p}{\varepsilon \delta} \right)$$

We can now apply the replicable rounding algorithm (cf. Algorithm D.4) to achieve the desired outcome. Indeed, by Theorem D.20, the output $\Lambda$ after the rounding procedure is $\rho$-reproducible and offsets the average $\bar{\Xi}$ by at most $\varepsilon$. Hence we have

$$\Lambda \in \left[ \frac{\text{OPT}}{\beta} - 2\varepsilon, \text{OPT} + 2\varepsilon. \right]$$

with probability at least $1 - \delta$. Finally, the algorithm has sample complexity

$$O\left( \frac{k^2 d^2}{\varepsilon^6 \rho^6} \log^2 \frac{p}{\varepsilon \rho \delta} \right)$$

Choosing $\varepsilon' = \varepsilon/2$ completes the proof. $\qquad\square$

**Theorem D.23** (OPT Estimation with Relative Error). *Fix $\varepsilon, \rho \in (0,1)$ and $\delta \in (0, \rho/3)$. There is a $\rho$-replicable algorithm such that with probability at least $1 - \delta$, it outputs an estimate $\Lambda$ of $\mathrm{OPT}$ where*

$$\Lambda \in \left[ \frac{1}{\beta(1+\varepsilon)} \, \mathrm{OPT}, (1+\varepsilon) \, \mathrm{OPT} \right].$$

*Moreover, it has sample complexity*

$$\tilde{O} \left( \frac{k^2 d^2 \beta^{12}}{\varepsilon^{12} \rho^6 \cdot \mathrm{OPT}^{12}} \log \frac{\rho}{\delta} \right).$$

*Proof.* We run the estimation algorithm with additive error with

$$\varepsilon_i := 2^{-i}$$
$$\rho_i := 2^{-i} \rho$$
$$\delta_i := 2^{-i} \delta$$

for $i = 1, 2, 3, \ldots$ until we obtain an estimate $\Lambda_i$ such that $\varepsilon_i \leq \varepsilon \Lambda_i / 2$.

Remark that

$$2^{-i} \leq \frac{1}{2} \varepsilon \Lambda_i$$
$$\Longleftarrow \quad 2^{-i} \leq \frac{1}{2} \varepsilon \left( \frac{\mathrm{OPT}}{\beta} - 2^{-i} \right)$$
$$\Longleftrightarrow \quad 2^{-i} \left( 1 + \frac{1}{2} \varepsilon \right) \leq \frac{\varepsilon \, \mathrm{OPT}}{2\beta}$$
$$\Longleftarrow \quad -i \leq \log \frac{\varepsilon \, \mathrm{OPT}}{4\beta}$$
$$\Longleftrightarrow \quad i = \log \frac{\beta}{\varepsilon \, \mathrm{OPT}} + 2.$$

Thus the algorithm certainly terminates.

Upon termination, we output some $\Lambda$ such that $\Lambda \in [\mathrm{OPT}/\beta - \varepsilon \Lambda/2, \mathrm{OPT} + \varepsilon \Lambda/2]$. It follows that

$$\Lambda \leq \mathrm{OPT} + \frac{1}{2} \varepsilon \Lambda$$
$$\left( 1 - \frac{1}{2} \varepsilon \right) \Lambda \leq \mathrm{OPT}$$
$$\Lambda \leq \frac{1}{1 - \frac{\varepsilon}{2}} \, \mathrm{OPT}$$
$$\leq (1 + \varepsilon) \, \mathrm{OPT}$$

and

$$\Lambda \geq \frac{\mathrm{OPT}}{\beta} - \frac{1}{2} \varepsilon \Lambda$$
$$\beta \left( 1 + \frac{1}{2} \varepsilon \right) \Lambda \geq \mathrm{OPT}$$
$$\Lambda \geq \frac{1}{\beta(1+\varepsilon)} \, \mathrm{OPT}.$$

The probability of not being replicable in each iteration is $2^{-i} \rho$. Hence the total probability is at most

$$\sum_{i=1}^{\infty} 2^{-i} \rho = \rho$$

and similarly for $\delta$.

Finally, there are $O\left(\log \frac{\beta}{\varepsilon \text{OPT}}\right)$ iterations and the sample complexity of each iteration is bounded above by the sample complexity of the final iteration where

$$2^{-i} = \Omega\left(\frac{\varepsilon\,\text{OPT}}{\beta}\right).$$

Hence the total sample complexity is

$$\tilde{O}\left(\log\frac{\beta}{\varepsilon\,\text{OPT}} \cdot \frac{k^2 d^2}{\left(\frac{\varepsilon\,\text{OPT}}{\beta}\right)^6 \left(\frac{\varepsilon\,\text{OPT}}{\beta}\cdot\rho\right)^6} \log\frac{p}{\delta}\right) = \tilde{O}\left(\frac{k^2 d^2 \beta^{12}}{\varepsilon^{12}\rho^6 \cdot \text{OPT}^{12}}\log\frac{p}{\delta}\right). \qquad \square$$

## D.6   Putting it Together

We are now ready to prove the results on statistical $k$-medians and statistical $k$-means, which we restate below for convenience. Recall that $\beta$ is an absolute constant and hence disappears under the big-O notation.

**Theorem 4.2** (Theorem 3.1; Formal). *Let $\varepsilon, \rho \in (0, 1)$ and $\delta \in (0, \rho/3)$. Given black-box access to a $\beta$-approximation oracle for weighted $k$-medians (cf. Problem 2.2), there is a $\rho$-replicable algorithm for statistical $k$-medians (cf. Problem 2.1) such that, with probability at least $1 - \delta$, it outputs a $(1+\varepsilon)\beta$-approximation. Moreover, it has sample complexity*

$$\tilde{O}\left(\left(\frac{k^2 d^2}{\varepsilon^{12}\rho^6 \cdot \text{OPT}^{12}} + \frac{k^3 2^{18d}\Delta^{3d+3}}{\rho^2 \varepsilon^{3d+5} \cdot \text{OPT}^3}\right)\log\frac{1}{\delta}\right).$$

*Proof (Theorem 4.2).* First, we compute $\Lambda$, an estimate of OPT with relative error $\varepsilon$, confidence $\delta/3$, and replicability parameter $\rho/3$. Then, we produce an $\varepsilon/2$-coreset for $k$-medians of cardinality $N$ with confidence and replicability parameters $\delta/3, \rho/3$. Finally, we estimate the mass at each point of the coreset with confidence and replicability parameters $\delta/3, \rho/3$ to an accuracy of

$$\frac{\varepsilon\Lambda}{4N} \le \frac{\varepsilon\,\text{OPT}}{2N}.$$

By Proposition D.19, running the approximation oracle on the coreset with the estimated weights yields a $\beta(1+\varepsilon)$-approximation.

By Theorem D.23, computing $\Lambda$ requires

$$\tilde{O}\left(\frac{k^2 d^2}{\varepsilon^{12}\rho^6 \cdot \text{OPT}^{12}}\log\frac{1}{\delta}\right)$$

samples.

By Theorem D.16, the coreset construction yields a coreset of cardinality

$$N = O\left(\frac{k 2^{5d}\Delta^{d+1}}{\varepsilon^{d+1}}\log\frac{\Delta}{\varepsilon \cdot \text{OPT}} \cdot 2^d\right) = O\left(\frac{k 2^{6d}\Delta^{d+1}}{\varepsilon^{d+1}}\log\frac{\Delta}{\varepsilon \cdot \text{OPT}}\right)$$

and incurs a sample cost of

$$\tilde{O}\left(\frac{k^3 2^{15d}\Delta^{3d+3}}{\rho^2 \varepsilon^{3d+3} \cdot \text{OPT}^3} \cdot 2^{3d}\frac{1}{\delta}\right) = \tilde{O}\left(\frac{k^3 2^{18d}\Delta^{3d+3}}{\rho^2 \varepsilon^{3d+3} \cdot \text{OPT}^3}\frac{1}{\delta}\right).$$

Finally, Corollary D.21 states that the probability mass estimation incurs a sample cost of

$$\tilde{O}\left(\frac{N^3}{\varepsilon^2\rho^2 \cdot \Lambda^2}\log\frac{1}{\delta}\right) = \tilde{O}\left(\frac{N^3}{\varepsilon^2\rho^2 \cdot \text{OPT}^2}\log\frac{1}{\delta}\right)$$

$$= \tilde{O}\left(\frac{k^3 2^{18d}\Delta^{3d+3}}{\varepsilon^{3d+3}} \cdot \frac{1}{\varepsilon^2\rho^2 \cdot \text{OPT}^2}\log\frac{1}{\delta}\right)$$

$$= \tilde{O}\left(\frac{k^3 2^{18d}\Delta^{3d+3}}{\rho^2 \varepsilon^{3d+5} \cdot \text{OPT}^2}\log\frac{1}{\delta}\right). \qquad \square$$

**Theorem 4.3** (Theorem 3.1; Formal)**.** *Given black-box access to a $\beta$-approximation oracle for weighted $k$-means (cf. Problem 2.2), there is a $\rho$-replicable algorithm for statistical $k$-means (cf. Problem 2.1) such that, with probability at least $1 - \delta$, it replicably outputs a $(1 + \varepsilon)\beta$-approximation. Moreover, it has sample complexity*

$$\tilde{O}\left(\left(\frac{k^2 d^2}{\varepsilon^{12}\rho^6 \cdot \mathrm{OPT}^{12}} + \frac{k^3 2^{39d}\Delta^{3d+6}}{\rho^2 \varepsilon^{6d+8} \cdot \mathrm{OPT}^3}\right)\log\frac{1}{\delta}\right) .$$

*Proof (Theorem 4.3).* Similar to $k$-median, we first compute $\Lambda$, an estimate of OPT with relative error $\varepsilon$, confidence $\delta/3$, and replicability parameter $\rho/3$. Then, we produce an $\varepsilon/2$-coreset for $k$-medians of cardinality $N$ with confidence and replicability parameters $\delta/3, \rho/3$. Finally, we estimate the mass at each point of the coreset with confidence and replicability parameters $\delta/3, \rho/3$ to an accuracy of

$$\frac{\varepsilon\Lambda}{4N} \le \frac{\varepsilon\,\mathrm{OPT}}{2N}.$$

By Proposition D.19, running the approximation oracle on the coreset with the estimated weights yields a $\beta(1 + \varepsilon)$-approximation.

By Theorem D.23, computing $\Lambda$ requires

$$\tilde{O}\left(\frac{k^2 d^2}{\varepsilon^{12}\rho^6 \cdot \mathrm{OPT}^{12}}\log\frac{1}{\delta}\right)$$

samples.

By Theorem D.18, the coreset construction yields a coreset of cardinality

$$N = O\left(\frac{k2^{11d}\Delta^{d+2}}{\varepsilon^{2d+2}}\log\frac{\Delta}{\varepsilon\,\mathrm{OPT}}\cdot 2^{2d}\right) = O\left(\frac{k2^{13d}\Delta^{d+2}}{\varepsilon^{2d+2}}\log\frac{\Delta}{\varepsilon\,\mathrm{OPT}}\right).$$

and incurs a sample cost of

$$\tilde{O}\left(\frac{k^3 2^{33d}\Delta^{3d+6}}{\rho^2\varepsilon^{6d+6}\cdot\mathrm{OPT}^3}\cdot 2^{6d}\right) = \tilde{O}\left(\frac{k^3 2^{39d}\Delta^{3d+6}}{\rho^2\varepsilon^{6d+6}\cdot\mathrm{OPT}^3}\frac{1}{\delta}\right).$$

Finally, Corollary D.21 states that the probability mass estimation incurs a sample cost of

$$\begin{aligned}
\tilde{O}\left(\frac{N^3}{\varepsilon^2\rho^2\cdot\Lambda^2}\frac{1}{\delta}\right) &= \tilde{O}\left(\frac{N^3}{\varepsilon^2\rho^2\cdot\mathrm{OPT}^2}\frac{1}{\delta}\right)\\
&= \tilde{O}\left(\frac{k^3 2^{39d}\Delta^{3d+6}}{\varepsilon^{6d+6}}\cdot\frac{1}{\varepsilon^2\rho^2\cdot\mathrm{OPT}^2}\frac{1}{\delta}\right)\\
&= \tilde{O}\left(\frac{k^3 2^{39d}\Delta^{3d+6}}{\rho^2\varepsilon^{6d+8}\cdot\mathrm{OPT}^2}\frac{1}{\delta}\right). \qquad\qquad \square
\end{aligned}$$

# E  The Euclidean Metric, Dimensionality Reduction, and $(k, p)$-Clustering

We now build towards a proof for Theorem 5.1, which states the formal guarantees of the replicable $k$-medians and $k$-means algorithms under the Euclidean distance. Let us begin by recalling the Johnson-Lindenstrauss lemma.

**Theorem E.1** ([Johnson, 1984])**.** *There exists a family of random linear maps $\pi_{d,m}$ from $\mathbb{R}^d \to \mathbb{R}^m$ such that the following hold.*

(i) *For every $d \ge 1, \varepsilon, \delta \in (0, 1/2), x \in \mathbb{R}^d$, and $m = O\left(1/\varepsilon^2 \log 1/\delta\right)$,*

$$\frac{1}{1 + \varepsilon}\|\pi x\|_2 \le \|x\|_2 \le (1 + \varepsilon)\|\pi x\|_2$$

*with probability at least $1 - \delta$.*

(ii) *$\pi_{d,m}$ is sub-Gaussian-tailed, thus for every unit vector $x \in \mathbb{R}^d$ and $\varepsilon > 0$,*

$$\|\pi x\|_2 \ge 1 + \varepsilon$$

*with probability at most $\exp\left(-\Omega(\varepsilon^2 m)\right)$.*

*Furthermore, we can take $\pi_{d,m}$ to be the set of random orthogonal projections $\mathbb{R}^d \to \mathbb{R}^m$ scaled by a factor of $\sqrt{d/m}$.*

We say that $\pi \in \pi_{d,m}$ is a *standard random dimension-reduction map* and write $\mathbb{P}_\pi$ to denote the projected distribution on $\mathbb{R}^m$, i.e., $\mathbb{P}_\pi(\cdot) = \mathbb{P}(\pi^{-1}(\cdot))$.

As mentioned in Section 5, it is not entirely clear how to translate between sets of centers for high-dimensional data and solutions sets for the compressed data. For this reason, our goal is to produce a clustering function $f : \mathcal{X} \to [k]$ which computes the label for any given point in polynomial time. This also requires us to consider another notion of cost which is translated more easily.

**Definition E.2** (Partition Cost). Define the cost of a $k$-partition $\mathcal{C} = (C^{(1)}, \ldots, C^{(k)})$ of $\mathcal{X}$ as

$$\mathrm{cost}(\mathcal{C}) := \min_{u^{(j)} \in \mathbb{R}^d : j \in [k]} \mathbb{E}_x \left[ \sum_{j=1}^k \mathbb{1}\{x \in C^{(j)}\} \cdot \kappa(x, u^{(j)})^p \right].$$

i.e., we consider the expected cost of picking the optimal centers w.r.t. the underlying data-generating distribution. Similarly, we define the following sample partition cost.

**Definition E.3** (Sample Partition Cost). We define the sample cost of a $k$-partition $\mathcal{C}$ of $x_1, \ldots, x_n$ as

$$\widehat{\mathrm{cost}}(\mathcal{C}, w) = \min_{u^{(j)} \in \mathbb{R}^d : j \in [k]} \sum_{i=1}^n w_i \sum_{j=1}^k \mathbb{1}\{x_i \in C^{(j)}\} \kappa(x_i, u^{(j)})^p$$

where $w \geq 0$, $\sum_i w_i = 1$. i.e., we consider the cost that is induced by the distribution specified by $w$ on the samples. We write $\widehat{\mathrm{cost}}(\mathcal{C})$ to denote the cost with the uniform distribution on the samples.

Remark that if $\mathcal{C}$ is induced by $F$, then $\mathrm{OPT} \leq \mathrm{cost}(\mathcal{C}) \leq \mathrm{cost}(F)$ and similarly for the sample partition cost as well.

**Theorem E.4** ([Makarychev et al., 2019]). *Let $x_1, \ldots, x_n \in \mathbb{R}^d$ be an instance of the Euclidean $(k, p)$-clustering problem. Fix $\varepsilon \in (0, 1/4), \delta \in (0, 1)$ and suppose $\pi : \mathbb{R}^d \to \mathbb{R}^m$ is a standard random dimension-reduction map with*

$$m = O\left( \frac{p^4}{\varepsilon^2} \log \frac{k}{\varepsilon\delta} \right).$$

*Then with probability at least $1 - \delta$,*

$$\frac{1}{1 + \varepsilon} \widehat{\mathrm{cost}}(\mathcal{C}) \leq \widehat{\mathrm{cost}}(\pi(\mathcal{C})) \leq (1 + \varepsilon)\widehat{\mathrm{cost}}(\mathcal{C})$$

*for every $k$-partition $\mathcal{C}$ of $x_1, \ldots, x_n$.*

In essence, this theorem tells us that if we let the centers of the clusters be the centers of the partition, then the (sample) cost is preserved in the lower dimensional space. To the best of our knowledge, this is the strongest result for dimensionality reduction in (sample) $(k, p)$-clustering which preserves more than just the costs of optimal centers.

The rest of Appendix E serves to generalize Theorem E.4 for preserving partition costs in the distributional setting. First, Appendix E.1 addresses the issue that a random orthogonal projection can increase the diameter of our data in the worst case by scaling the original data by an appropriate factor. Next, Appendix E.2 slightly generalizes Theorem E.4 to the weighted sample clustering setting. Finally, Appendix E.3 extends Theorem E.4 to the distributional setting, culminating with Theorem E.10, which may be of independent interest beyond replicability.

Recall that $\mathbb{P}_\pi$ is the push-forward measure on $\pi(\mathcal{X})$ induced by $\pi$. We write $\mathrm{OPT}_\pi$ to denote the cost of an optimal solution to the statistical $(k, p)$-clustering problem on $(\pi(\mathcal{X}), \mathbb{P}_\pi)$ and $\widehat{\mathrm{OPT}}_\pi$ for the $(k, p)$-clustering problem on $\pi(x_1), \ldots, \pi(x_n)$.

### E.1 Scaling

We may assume that a standard dimension-reduction map $\pi$ satisfies $\pi = \sqrt{d/m}P$ for some orthogonal projection $P$. Hence $\pi$ has operator norm at most $\sqrt{d}$ and
$$\pi(\mathcal{X}) \subseteq \sqrt{d}\mathcal{B}_m.$$

In order to apply our existing analysis, we first scale down the samples in the input space by a factor of $\sqrt{d}$ so that the projected space still lives in a $\kappa$-ball of diameter 1. This does not affect our approximation guarantees as they are all multiplicative and hence scale-invariant. However, the analysis we perform is on the scaled distribution with optimal cost $\mathrm{OPT}' = d^{-p/2}\,\mathrm{OPT}$. This extra term must be accounted for.

We proceed assuming now that $\mathcal{X} \subseteq \frac{1}{\sqrt{d}}\mathcal{B}_d$ and $\pi(\mathcal{X}) \subseteq \mathcal{B}_m$.

### E.2 Preserving Partition Costs

Let $\widehat{\mathrm{cost}}(\mathcal{C}, w), \widehat{\mathrm{cost}}(\pi(\mathcal{C}), w)$ be the weighted cost of the partition and projected partition, respectively. Recall this means that the points participate with weights $w$ in the objective. Essentially, the next result shows that since the projection guarantees do not depend on the number of points we are projecting, we can simulate the weights by considering multiple copies of the points.

**Proposition E.5.** *Let $x_1, \ldots, x_n \in \mathbb{R}^d$ be an instance of the Euclidean $(k, p)$-clustering problem and $w_1, \ldots, w_n \in \mathbb{R}_+$ be weights which sum to 1. Fix $\varepsilon \in (0, 1/4), \delta \in (0, 1)$, and suppose $\pi : \mathbb{R}^d \to \mathbb{R}^m$ is a standard random dimension-reduction map with*
$$m = O\left(\frac{p^4}{\varepsilon^2}\log\frac{k}{\varepsilon\delta}\right).$$

*Then with probability at least $1 - \delta$,*
$$\frac{1}{1+\varepsilon}\widehat{\mathrm{cost}}(\mathcal{C}, w) \leq \widehat{\mathrm{cost}}(\pi(\mathcal{C}), w) \leq (1+\varepsilon)\widehat{\mathrm{cost}}(\mathcal{C}, w),$$
*for every $k$-partition $\mathcal{C}$ of $x_1, \ldots, x_n$.*

*Proof (Proposition E.5).* First suppose that $w_i = a_i/b_i$ for $a_i, b_i \in \mathbb{Z}_+$ and consider
$$w_i' := \mathrm{lcm}(b_1, \ldots, b_n)w_i \in \mathbb{Z}_+$$
obtained from $w_i$ by multiplying by the least common multiple of the denominators.

Let $y_1, \ldots, y_{n'}$ be the multiset obtained from $x_1, \ldots, x_n$ by taking $w_i'$ multiples of $x_i$. The cost of any partition of $x_1, \ldots, x_n$ with weights $q_1, \ldots, q_n$ is equal to the cost of the induced partition of $y_1, \ldots, y_{n'}$ with uniform weights. Thus we can apply Theorem E.4 on $y_1, \ldots, y_{n'}$ to conclude the proof.

Now, for general $w_i \in \mathbb{R}_+$, we remark by the density of the rationals in $\mathbb{R}$ that there are some $q_1, \ldots, q_n \in \mathbb{Q}_+$ which sum to 1 and satisfy
$$\sum_{i=1}^{n}|q_i - w_i| < \frac{\varepsilon\min(\widehat{\mathrm{OPT}}, \widehat{\mathrm{OPT}}_\pi)}{2}.$$

Then for any $k$-partition $\mathcal{C}$ and $u^{(1)}, \ldots, u^{(k)} \in \mathcal{B}_d$,
$$\left|\sum_{i=1}^{n} q_i \sum_{j=1}^{k} \mathbb{1}\{x_i \in C^{(j)}\}\kappa(x_i, u^{(j)})^p - \sum_{i=1}^{n} w_i \sum_{j=1}^{k} \mathbb{1}\{x_i \in C^{(j)}\}\kappa(x_i, u^{(j)})^p\right|$$
$$\leq \sum_{i=1}^{n}|q_i - w_i| \sum_{j=1}^{k} \mathbb{1}\{x_i \in C^{(j)}\}\kappa(x_i, u^{(j)})^p$$
$$\leq \sum_{i=1}^{n}|q_i - w_i|$$
$$\leq \frac{\varepsilon}{2}\widehat{\mathrm{OPT}}.$$

In particular

$$\frac{1}{1+\varepsilon}\operatorname{cost}(\mathcal{C},w) \le \operatorname{cost}(\mathcal{C},q) \le (1+\varepsilon)\operatorname{cost}(\mathcal{C},w).$$

Similarly, for any $v^{(1)},\dots,v^{(k)} \in \mathcal{B}_m$,

$$\left| \sum_{i=1}^{n} q_i \sum_{j=1}^{k} \mathbb{1}\{x_i \in C^{(j)}\}\kappa(\pi(x_i),v^{(j)})^p - \sum_{i=1}^{n} w_i \sum_{j=1}^{k} \mathbb{1}\{x_i \in C^{(j)}\}\kappa(\pi(x_i),v^{(j)})^p \right|$$
$$\le \frac{\varepsilon}{2}\widehat{\operatorname{OPT}}_\pi,$$

which implies that

$$\frac{1}{1+\varepsilon}\operatorname{cost}(\pi(\mathcal{C}),w) \le \operatorname{cost}(\pi(\mathcal{C}),q) \le (1+\varepsilon)\operatorname{cost}(\pi(\mathcal{C}),w).$$

Finally, we conclude that

$$\begin{aligned}
\operatorname{cost}(\mathcal{C},w) &\le (1+\varepsilon)\operatorname{cost}(\mathcal{C},q) \\
&\le (1+\varepsilon)^2 \operatorname{cost}(\pi(\mathcal{C}),q) \\
&\le (1+\varepsilon)^3 \operatorname{cost}(\pi(\mathcal{C}),w) \\
\operatorname{cost}(\pi(\mathcal{C}),w) &\le (1+\varepsilon)^3 \operatorname{cost}(\mathcal{C},w).
\end{aligned}$$

Choosing $\varepsilon'$ to be a constant fraction of $\varepsilon$ concludes the proof. $\qquad\square$

**Corollary E.6.** *Let $x_1,\dots,x_n \in \mathbb{R}^d$ be an instance of the weighted Euclidean $(k,p)$-clustering problem with non-negative weights $w_1,\dots,w_n \in \mathbb{R}_+$ which sum to 1. Fix $\varepsilon \in (0,1/4), \delta \in (0,1)$ and suppose $\pi : \mathbb{R}^d \to \mathbb{R}^m$ is a standard random dimension-reduction map with*

$$m = O\left(\frac{p^4}{\varepsilon^2}\log\frac{k}{\varepsilon\delta}\right).$$

*Then with probability at least $1-\delta$,*

$$\frac{1}{1+\varepsilon}\widehat{\operatorname{OPT}} \le \widehat{\operatorname{OPT}}_\pi \le (1+\varepsilon)\widehat{\operatorname{OPT}}.$$

*Proof (Corollary E.6).* The optimal clustering cost coincides with the cost of the partition induced by the optimal centers. $\qquad\square$

### E.3 Dimensionality Reduction

Let $G = \{g^{(1)},\dots,g^{(k)}\}$ be a $\beta$-approximate solution to the statistical Euclidean $(k,p)$-clustering problem on $(\pi(\mathcal{X}),\mathbb{P}_\pi)$. We would like to argue that the partition of $\mathcal{X}$ induced by $G$ is a $(1+\varepsilon)\beta$-approximate partition with probability at least $1-\delta$. To do so, we first approximate $\mathcal{X}$ with a finite set $\tilde{\mathcal{X}}$ in a replicable fashion while preserving partition costs. This allows us to apply Proposition E.5.

Recall we write $\Lambda$ to denote a replicable estimate of OPT such that $\Lambda \in [\mathrm{OPT}/\beta(1+\varepsilon), (1+\varepsilon)\,\mathrm{OPT}]$ (cf. Theorem D.23). Consider a grid of length $\varepsilon\Lambda/(4p\Delta)$. for a point $x \in \mathcal{X}$, we assign it to the center of the cell that it belongs to, say $\tilde{x}$. Let $\tilde{\mathcal{X}} := \{\tilde{x} : x \in \mathcal{X}\}$ denote this finite set and $\tilde{\mathbb{P}}$ the distribution induced by the discretization. Remark that $\kappa(x,\tilde{x}) \le \varepsilon\Lambda/(4p)$ for every $x \in \mathcal{X}$.

Note that a partition of $\mathcal{X}$ induces a partition of $\tilde{\mathcal{X}}$ and vice versa. We proceed without making any distinction between the two.

**Proposition E.7.** *For any point $u \in \mathcal{B}_d$,*

$$|\kappa(x,u)^p - \kappa(\tilde{x},u)^p| \le p|\kappa(x,u) - \kappa(\tilde{x},u)|.$$

*Proof.* By the mean value theorem, the function $g : [0,1] \to \mathbb{R}$ given by $z \mapsto z^p$ is $p$-Lipschitz:

$$\begin{aligned}
|g(y) - g(z)| &\le \sup_{\xi\in[0,1]} g'(\xi)|y-z| \\
&\le p|y-z|. \qquad\qquad\square
\end{aligned}$$

We write $\widetilde{\mathrm{cost}}(\mathcal{C})$ to denote the cost of the partition $\mathcal{C}$ of $\tilde{\mathcal{X}}$.

**Proposition E.8.** *For any partition $\mathcal{C}$ of $\mathcal{X}$*

$$\frac{1}{1+\varepsilon}\,\mathrm{cost}(\mathcal{C}) \leq \widetilde{\mathrm{cost}}(\mathcal{C}) \leq (1+\varepsilon)\,\mathrm{cost}(\mathcal{C}).$$

*Proof.* We have

$$\left| \mathbb{E}_x \left[ \sum_{j=1}^k \mathbb{1}\{x \in C^{(j)}\} \cdot \kappa(x, u^{(j)})^p \right] - \mathbb{E}_x \left[ \sum_{j=1}^k \mathbb{1}\{x \in C^{(j)}\} \cdot \kappa(\tilde{x}, u^{(j)})^p \right] \right|$$

$$\leq \mathbb{E}_x \left[ \sum_{j=1}^k \mathbb{1}\{x \in C^{(j)}\} \cdot |\kappa(x, u^{(j)})^p - \kappa(\tilde{x}, u^{(j)})^p| \right]$$

$$\leq \mathbb{E}_x \left[ \sum_{j=1}^k \mathbb{1}\{x \in C^{(j)}\} \cdot p|\kappa(x, u^{(j)}) - \kappa(\tilde{x}, u^{(j)})| \right]$$

$$\leq p \cdot \frac{\varepsilon \Lambda}{4p}$$

$$\leq \frac{\varepsilon}{2}\,\mathrm{OPT}. \qquad \square$$

**Corollary E.9.** *Fix $\varepsilon \in (0, 1/4), \delta \in (0, 1)$ and suppose $\pi : \mathbb{R}^d \to \mathbb{R}^m$ is a standard random dimension-reduction map with*

$$d = O\left( \frac{p^4}{\varepsilon^2} \log \frac{k}{\varepsilon\delta} \right).$$

*We have*

$$\frac{1}{(1+\varepsilon)}\,\mathrm{OPT} \leq \widetilde{\mathrm{OPT}}_\pi \leq (1+\varepsilon)\,\mathrm{OPT},$$

*with probability at least $1 - \delta$, where the probability is taken w.r.t. the choice of the random map.*

*Proof.* The optimal clustering cost coincides with the cost of the partition induced by the optimal centers. An application of Proposition E.8 yields the desired result. $\qquad \square$

**Theorem E.10.** *Fix $\varepsilon \in (0, 1/4), \delta \in (0, 1)$ and suppose $\pi \in \pi_{d,m}$ is a standard random dimension-reduction map with*

$$m = O\left( \frac{p^4}{\varepsilon^2} \log \frac{k}{\varepsilon\delta} \right).$$

*Let $G = \{g^{(1)}, \ldots, g^{(k)}\}$ be a $\beta$-approximate solution to the statistical Euclidean $(k, p)$-clustering problem on $(\pi(\tilde{\mathcal{X}}), \tilde{\mathbb{P}}_\pi)$. Then the partition of $\mathcal{X}$ induced by $G$ is a $(1+\varepsilon)\beta$-approximate partition with probability at least $1 - \delta$.*

*That is, for $\mathcal{C} = \{C^{(1)}, \ldots, C^{(k)}\}$ given by*

$$C^{(j)} := \left\{ x \in \mathcal{X} : g^{(j)} = \mathrm{argmin}_{g \in G}\, \kappa(\pi(\tilde{x}), g^{(j)})^p \right\},$$

*we have*

$$\mathrm{cost}(\mathcal{C}) \leq (1+\varepsilon)\beta\,\mathrm{OPT}.$$

*Proof.* We have

$$\begin{aligned}
\mathrm{cost}(\mathcal{C}) &\leq (1+\varepsilon)\widetilde{\mathrm{cost}}(\mathcal{C}) && \text{Proposition E.8} \\
&\leq (1+\varepsilon)^2\widetilde{\mathrm{cost}}(\pi(\mathcal{C})) && \text{Proposition E.5} \\
&\leq (1+\varepsilon)^2\widetilde{\mathrm{cost}}(G) && G \text{ induces } \pi(\mathcal{C}) \\
&\leq (1+\varepsilon)^2\beta\widetilde{\mathrm{OPT}}_\pi && G \text{ is } \beta\text{-approximate} \\
&\leq (1+\varepsilon)^3\beta\,\mathrm{OPT}. && \text{Corollary E.9}
\end{aligned}$$

Choosing $\varepsilon'$ to be a constant fraction of $\varepsilon$ concludes the proof. $\qquad \square$

### E.4 Summary of Dimensionality Reduction

Before proving Theorem 5.1, let us summarize the stages of our pipeline.

1) Scale the input data by $1/\sqrt{d}$.
2) Estimate OPT of the scaled data within a multiplicative error.
3) Discretize the scaled data with a fine grid.
4) Project the scaled data with a random map $\pi \in \pi_{d,m}$.
5) Compute a coreset on the projected data.
6) Estimate the probability mass at each point of the coreset.
7) Call the $\beta$-approximation oracle to produce $k$ centers $g^{(1)}, \ldots, g^{(k)}$.
8) Output the clustering function induced by the centers.

The clustering function proceeds as follows. Given a point $x \in \mathcal{X}$,

1) Scale $x$ by $1/\sqrt{d}$.
2) Discretize the scaled point with the same grid used in the algorithm.
3) Project the discretized point using the same $\pi$ used in the algorithm.
4) Output the label of the closest center $g^{(j)}$ in the lower dimension for $j \in [k]$.

We are now ready to prove Theorem 5.1, which we restate below for convenience.

**Theorem 5.1** (Theorem 3.2; Formal). *Let $\varepsilon, \rho \in (0,1)$ and $\delta \in (0, \rho/3)$. Given a $\beta$-approximation oracle for weighted Euclidean $k$-medians ($k$-means), there is a $\rho$-replicable algorithm that outputs a clustering function such that with probability at least $1 - \delta$, the cost of the partition is at most $(1 + \varepsilon)\beta \operatorname{OPT}$. Moreover, the algorithm has sample complexity*

$$\tilde{O}\left(\operatorname{poly}\left(\frac{kd}{\rho\operatorname{OPT}}\right)\left(\frac{2\sqrt{m}}{\varepsilon}\right)^{O(m)}\log\frac{1}{\delta}\right),$$

*where $m = O\left(\frac{1}{\varepsilon^2}\log\frac{k}{\delta\varepsilon}\right)$.*

*Proof (Theorem 5.1).* We randomly draw some $\pi \in \pi_{d,m}$ for

$$d = O\left(\frac{p^4}{\varepsilon^2}\log\frac{k}{\varepsilon\delta}\right).$$

Then, we discretize a sufficiently large sample after scaling and push it through $\pi$. We then run the corresponding algorithm (cf. Theorem 4.2, Theorem 4.3) to produce some centers $G$ for the compressed data. This allows us to partition $\mathcal{X}$ according to the closest member of $G$ in the lower dimension. An application of Theorem E.10 yields the performance guarantees of this algorithm.

The sample complexity guarantee from Theorem 4.2, Theorem 4.3 is for the scaled distribution and the scaled optimal cost satisfies $\operatorname{OPT}'_\pi = d^{-p/2}\operatorname{OPT}_\pi$. Hence the sample complexity resolves to

$$\tilde{O}\left(\operatorname{poly}\left(\frac{kd}{\rho\operatorname{OPT}}\right)\left(\frac{2\sqrt{m}}{\varepsilon}\right)^{O(m)}\log\frac{1}{\delta}\right)$$

We conclude the proof by remarking that for the Euclidean metric, $\Delta = \sqrt{d}$ and that $O(\log n)^{O(\log n)} = n^{O(\log\log n)}$. $\qquad\square$

## F $\quad k$-Centers

In this section, we will explain our approach to solving the statistical $k$-centers problem (cf. Problem 2.3). As we explained before, since this problem has a different flavor compared to $(k, p)$-clustering, we introduce some extra assumptions in order to be able to solve it from samples. To the best of our knowledge, this is the first work that considers a model of statistical flavor for the $k$-centers problem.

## F.1 Assumptions

First, we describe an assumption that is necessary to obtain good clustering solutions even if we do not insist on using replicable algorithms.

**Assumption F.1** (Clusterable). For some absolute constants $\beta, B > 0$, an i.i.d. sample $S = x_1, \ldots, x_n$ of size $n$ from $\mathbb{P}$ has the following property. With probability at least $q$, there exists some $(\beta, B)$-approximate solution $F_*$ to Problem 2.3 such that, for every $f \in F_*$ there is some $x_f \in S$ with $F_*(x_f) = f$. Here $q \in [0, 1]$.

Assumption F.1 is necessary to obtain good clustering solutions even in the non-replicable case. Our distribution $\mathbb{P}$ must be sufficiently well-behaved so that solving the sample $k$-centers problem translates to a good solution in the population case. Essentially, this assumption states that with high probability, the sample will contain a point from every single cluster of *some* $(\beta, B)$-approximate solution. In order to design *replicable* algorithms, we need to make a stronger assumption.

**Assumption F.2** (Replicable). For some absolute constants $\beta, B$, there exists a $(\beta, B)$-approximate solution $F_*$ to Problem 2.3 with the following property. With probability at least $q$, in an i.i.d. sample $S = x_1, \ldots, x_n$ of size $n$ from $\mathbb{P}$, for every $f \in F_*$, there is some $x_f \in S$ with $F_*(x_f) = f$. Here $q \in [0, 1]$.

Note that Assumption F.1 guarantees that, with high probability, every sample will help us identify *some* good solution. However, these good solutions could be vastly different when we observe different samples, thus this assumption is not sufficient to design a replicable algorithm with high utility. Assumption F.2 is necessary to attain replicable solutions. Indeed, this assumption essentially states that there is a *fixed* solution for which we will observe samples from every cluster with high probability. If this does not hold and we observe points from clusters of solutions that vary significantly, we cannot hope to replicably recover a fixed approximate solution. Note that Assumption F.2 is stronger than Assumption F.1, since we have flipped the order of the existential quantifiers. One way to satisfy this assumption is to require that there is some $(\beta, B)$-approximate solution so that the total mass that is assigned to every cluster of it is at least $1/n$ for some $n \in \mathbb{N}$. Then, by observing $\widetilde{O}(n)$ points, we will receive at least one from every cluster with high probability. This is made formal in Proposition F.3.

**Proposition F.3** ([Janson, 2018]). *Let $\delta \in (0, 1)$. Under Assumption F.2, a sample of*

$$O\left(\frac{nm}{q} \log \frac{1}{\delta}\right)$$

*points contains at least $m$ points from each cluster of $F_*$ with probability at least $1 - \delta$.*

*Proof (Proposition F.3).* Let $Y_i$ be a geometric variable with success rate $q$, which denotes the number of batches of $n$ samples we draw until a point from each cluster of $F_*$ is observed. Then $Y := \sum_{i=1}^{m} Y_i$ upper bounds the number of batches until we obtain $m$ points from each cluster.

As shown in Janson [2018],

$$\mathbb{P}\{Y \geq \lambda \mathbb{E}Y\} \leq \exp(1 - \lambda)$$
$$\mathbb{E}Y = \frac{m}{q}.$$

The result follows by picking $\lambda = 1 + \log(1/\delta)$. $\qquad\square$

## F.2 High-Level Overview of the Approach

We are now ready to provide a high-level overview of our approach to derive Theorem 3.3. We first take a fixed grid of side $c$ in order to cover the unit ball. Then, we sample sufficiently many points from $\mathbb{P}$. Subsequently, we round all the points of the sample to the centers of the cells of the grid that they fall into. Using these points, we empirically estimate the density of the distribution on every cell of the grid. Next, in order to ensure that our solution is replicable, we take a random threshold from a predefined interval and discard the points from all the cells whose density falls below the threshold. Finally, we call the approximation oracle that we have access to using the points that have survived after the previous step. We remark that unlike the $(k, p)$-clustering problem (cf. Problem 2.1), to the best of our knowledge, there does not exist any dimensionality reduction techniques that apply to the $k$-centers problem. In the following subsections we explain our approach in more detail.

### F.3 Oracle on Sample

As we explained before, we require black-box access to an oracle $\mathcal{O}$ for the combinatorial $k$-centers problem on a sample which outputs $(\widehat{\beta}, \widehat{B})$-approximate solutions $\widehat{F}$ for Problem 2.4. Thus, we need to show that, with high probability, the output of this solution is a good approximation to Problem 2.3. This is shown in Proposition F.4.

**Proposition F.4.** *Suppose there is a $(\beta, B)$-approximate solution $F_*$ for Problem 2.3 and that we are provided with a sample of size $O(n\log(1/\delta)/q)$ containing an observation from each cluster of $F_*$, with probability at least $1 - \delta$. Then, the output $\widehat{F}$ of the oracle $\mathcal{O}$ on the sample satisfies*

$$\kappa(x, \widehat{F}) \leq (2\beta + \widehat{\beta}) \operatorname{OPT} + 2B + \widehat{B}.$$

*for all $x \in \mathcal{X}$, with probability at least $1 - \delta$.*

In the previous result $(\beta, B), (\widehat{\beta}, \widehat{B})$ are the approximation parameters that we inherit due to the fact that we receive samples from $\mathbb{P}$ and that we solve the combinatorial problem approximately using $\mathcal{O}$, respectively.

*Proof (Proposition F.4).* Let $F_*$ denote the $(\beta, B)$-approximate solution from Assumption F.2. We condition on the event that such a solution exists in the sample, which occurs with probability at least $1 - \delta$.

Fix $x \in \mathcal{X}$, which is not necessarily observed in the sample. We pay a cost of $\beta \operatorname{OPT} + B$ to travel to the nearest center of $F_*$. Next, we pay another $\beta \operatorname{OPT} + B$ to travel to the observed sample from this center. Finally, we pay a cost of

$$\widehat{\operatorname{cost}}(\widehat{F}) \leq \widehat{\beta}\widehat{\operatorname{OPT}} + \widehat{B}$$
$$\leq \widehat{\beta}\operatorname{OPT} + \widehat{B}.$$

to arrive at the closest center in $\widehat{F}$. $\qquad\qquad\square$

### F.4 Oracle with Grid

The issue with using directly the output of $\mathcal{O}$ on a given sample is that the result will not be replicable, since we have no control over its behavior. In a similar way as with the $(k, p)$-clustering problem, the main tool that we have in our disposal to "stabilize" the output of $\mathcal{O}$ is to use a grid. Unlike the other problems, we cannot use the hierarchical grid approach in this one. The reason is that, due to the min-max nature of k-centers, we need to cover the whole space and not just the points where the distribution is dense. Recall that so far we have established that the oracle $\mathcal{O}$ outputs $(2\beta + \widehat{\beta}, 2B + 2\widehat{B})$-approximate solutions $F$ for Problem 2.3, given that Assumption F.2 holds. Algorithm F.1 "discretizes" the points of the sample using a grid with side-length $c$ before applying the oracle. Proposition F.5 shows that by doing that, we have to pay an extra additive term that is of the order $O(c\Delta)$.

---

**Algorithm F.1** Oracle with Grid

1: **Oracle with Grid**(oracle $\mathcal{O}$, sample $x_1, \ldots, x_n$, grid length $c$):
2: **for** $i \leftarrow 1, \ldots, n$ **do**
3:     Decompose $x_i = z_i \cdot c + x_i'$ for some $z_i \in \mathbb{Z}^d, x_i' \in [0, c)^d$
4:     $\tilde{x}_i \leftarrow z_i \cdot c + (c/2, c/2, \ldots, c/2)$ {$\tilde{x}_i$ is the center of the cell that contains $x_i$.}
5: **end for**
6: Return $\mathcal{O}(\tilde{x}_1, \ldots, \tilde{x}_n)$

---

**Proposition F.5.** *Suppose there is a $(\beta, B)$-approximate solution $F_*$ for Problem 2.3 and that we are provided with a sample of size $n$ containing an observation from each cluster of $F_*$. Let $\tilde{G}$ denote the output of Algorithm F.1 using a $(\hat{\beta}, \hat{B})-$approximation oracle $\mathcal{O}$ for Problem 2.4. Then, for any $x \in \mathcal{X}$,*

$$\kappa(x, \tilde{G}) \leq (2\beta + \widehat{\beta}) \operatorname{OPT} + 2B + \widehat{B} + (4\beta + 2\widehat{\beta} + 1)c\Delta.$$

*Proof (Proposition F.5).* For the sake of analysis, imagine we used the grid to discretize all points of $\mathcal{X} \mapsto \tilde{\mathcal{X}}$. Then $F_*$ is a solution with cost $\beta\widetilde{\mathrm{OPT}} + B + c\Delta$ and we observe a point from each of its clusters in our discretized sample. Here $\widetilde{\mathrm{OPT}}$ is the cost of a solution to Problem 2.3 on $\tilde{\mathcal{X}}$.

Proposition F.4 shows that the oracle returns a solution $\widetilde{G}$ with cost at most

$$(2\beta + \widehat{\beta})\widetilde{\mathrm{OPT}} + (2B + c\Delta) + \widehat{B}$$

on the discretized points.

Then for any $x \in \mathcal{X}$,

$$\begin{aligned}
\widetilde{\mathrm{OPT}} &= \kappa(x, \tilde{G}_{\mathrm{OPT}}) \\
&\leq \kappa(x, \tilde{x}) + \kappa(\tilde{x}, \tilde{G}_{\mathrm{OPT}}) \\
&\leq \kappa(x, \tilde{x}) + \kappa(\tilde{x}, F_{\mathrm{OPT}}) \\
&\leq 2\kappa(x, \tilde{x}) + \kappa(x, F_{\mathrm{OPT}}) \\
&\leq 2c\Delta + \mathrm{OPT}.
\end{aligned}$$

The conclusion follows by combining the two inequalities. $\qquad\square$

## F.5 Replicable Active Cells

At a first glance, we might be tempted to work towards replicability by arguing that, with high probability, every grid cell is non-empty in one execution of the algorithm if and only if it is non-empty in another execution. However, since we have no control over $\mathbb{P}$, it is not clear how one can prove such a statement. Our approach is to take more samples and ignore cells which do not have a sufficient number of points using some random threshold. The idea is similar to the heavy-hitters algorithm (cf. Algorithm D.2) originally conceived by Impagliazzo et al. [2022] and is presented in Algorithm F.2. Then, we move all the points of active cells to the center of the cell. Notice that in the $k$-center objective, unlike the $(k, p)$-clustering objective, we do not care about how much mass is placed in a cell, just whether it is positive or not.

We first derive a bound on the number of cells that the distribution puts mass on. It is not hard to see that since we are working in a bounded domain, the grid contains at most $(1/c)^d$ cells in total. Thus there are at most $M \leq (1/c)^d$ cells that have non-zero mass. Proposition F.6 shows how we can obtain such a bound in the unbounded domain setting.

**Proposition F.6.** *Let $F_*$ be a $(\beta, B)$-approximate solution for Problem 2.3. Let $M$ denote the maximum number of cells which intersect some cluster of $F_*$ and $\tilde{\mathbb{P}}$ the discretized distribution that is supported on centers of the cells of the grid of size $c$. Then, the support of $\tilde{\mathbb{P}}$ is at most $kM$ where*

$$M \leq \left( \frac{c + 2c\Delta + 2(\beta\,\mathrm{OPT} + B)}{c} \right)^d.$$

*Proof (Proposition F.6).* Fix a center $f$ and consider the cell $Z$ containing the center. The farthest point in the farthest cell in the cluster of $f$ is of distance at most $\beta\,\mathrm{OPT} + B + c\Delta$ and thus the cells we are concerned with sit in an $\ell_\infty$ ball of that radius. Not including $Z$, we walk past at most

$$\frac{\beta\,\mathrm{OPT} + B + c\Delta}{c}$$

cells along the canonical basis of $\mathbb{R}^d$, hence there are at most

$$\left[ 1 + 2c\Delta + \frac{2}{c}(\beta\,\mathrm{OPT} + B) \right]^d$$

such cells. $\qquad\square$

Notice that the previous discussion and Proposition F.5 reveal a natural trade-off in the choice of the parameter $c$: by reducing $c$, we decrease the additive error of our approximation algorithm but we increase the number of samples at a rate $(1/c)^d$. Since we have bounded the number of cells from which we can observe a point, the next step is to use a sufficiently large number of samples so that we can estimate replicably whether a cell is active or not. This is demonstrated in Algorithm F.2.

---

**Algorithm F.2** Replicable Active Cells

1: **rActiveCells**(grid length $c$):
2: $m \leftarrow \tilde{O}\left(\lambda k n M^2 / \rho^2\right)$
3: $N \leftarrow \lambda n m M$
4: Sample $x_1, \ldots, x_N \sim \mathbb{P}$,
5: Lazy initialize counter $Z = 0$ for every cell of the grid with length $c$
6: **for** $i \leftarrow 1, \ldots, N$ **do**
7: $\quad G_i \leftarrow$ cell that samples $x_i$ falls into
8: $\quad$ Increment $Z(G_i) \leftarrow Z(G_i) + 1$
9: **end for**
10: Choose $v \in [0, m/N]$ uniformly randomly
11: Output all $G_i$ such that $Z(G_i)/N \geq v$

---

**Proposition F.7.** *Suppose Assumption F.2 holds and let $F_*$ be the $(\beta, B)$-approximate solution. Let $\delta, \rho \in (0, 1)$. Then Algorithm F.2 is $\rho$-replicable and returns cells of the grid such that at least one sample from each cluster of $F_*$ falls into these cells with probability at least $1 - \delta$. Moreover, it has sample complexity*

$$\tilde{O}\left(\frac{n^2 k M^3 \log(1/\delta)}{q^2 \rho^2}\right).$$

*Proof (Proposition F.7).* Enumerate the $kM$ cells within the support of $\mathbb{P}$ and let the counter $Z^{(j)}$ denote the number of samples at the $j$-th cell. Then $Z = (Z^{(1)}, \ldots, Z^{(kM)})$ follows a multinomial distribution with parameters $(p^{(1)}, \ldots, p^{(kM)})$ and $N$, where $p$ is unknown to us. Nevertheless, by Proposition B.1, for any $\varepsilon \in (0, 1)$, sampling

$$N \geq \frac{\ln 5/\delta + kM \ln 2}{2\varepsilon^2}$$

points implies that $\sum_{j=1}^{a} |\widehat{p}^{(j)} - p^{(j)}| < 2\varepsilon$ with probability at least $1 - \delta/5$. From the definition of $N$, this is equivalent to

$$m \geq \frac{\ln 5/\delta + kM \ln 2}{nM\varepsilon^2}.$$

By Proposition F.3, we observe at least $mM$ points from each cluster of $F_*$ with probability at least $1 - \delta$.

Let $\widehat{p}_1^{(j)}, \widehat{p}_2^{(j)}$ be the empirical mean of $Z^{(j)}$ across two different executions of Algorithm F.2. Then by the triangle inequality, $\sum_{j=1}^{kM} |\widehat{p}_1^{(j)} - \widehat{p}_2^{(j)}| \leq 4\varepsilon$.

Since we observe at least $Mm$ points from each cluster of $F_*$, there is at least one point from each cluster with at least $m$ observations. Thus outputting the points $j$ such that $\widehat{p}^{(j)} \geq v$ for some $v \in [0, m/N]$ means we always output one point from each cluster.

Now, the outputs between two executions are not identical only if the point $v$ "splits" some pair $\widehat{p}_1^{(j)}, \widehat{p}_2^{(j)}$. This occurs with probability at most $4\varepsilon/(m/N) = 4N\varepsilon/m$, To bound this value by $\rho/5$, set

$$\varepsilon \leq \frac{\rho m}{20N}$$
$$= \frac{\rho}{20\lambda nM}.$$

Plugging this back yields

$$m \geq \frac{400\lambda nM \ln \frac{5}{\delta} + 400\lambda nkM^2 \ln 2}{\rho^2}.$$

All in all, Algorithm F.2 outputs the desired result with probability at least $1 - \delta$ and is $\rho$-replicable given

$$\tilde{O}\left(\frac{n^2 k M^3 \log(1/\delta)}{q^2 \rho^2}\right)$$

samples from $\mathbb{P}$. $\qquad\square$

### F.6 Putting it Together

We now have all the ingredients in place to prove Theorem 3.3. The proof follows directly by combining Proposition F.5, and Proposition F.7. Essentially, we replicably estimate the active cells of our grid and then move all the points of each active cell to its center.

For completeness, we also state the formal version of Theorem 3.3.

**Theorem F.8** (Theorem 3.3; Formal)**.** *Let $\delta, \rho \in (0, 1)$. Suppose Assumption F.2 holds and let $F_*$ be the $(\beta, B)$-approximate solution. Then, given black-box access to a $(\hat{\beta}, \hat{B})$-approximation oracle for Problem 2.4, Algorithm F.2 is $\rho$-replicable and returns a solution $\hat{F}$ such that*

$$\kappa(x, \hat{F}) \leq (2\beta + \widehat{\beta}) \, \mathrm{OPT} + 2B + \widehat{B} + (4\beta + 2\hat{\beta} + 1)c\Delta$$

*with probability at least $1 - \delta$. Moreover, the algorithm requires at most*

$$\tilde{O}\left(\frac{n^2 k \left(1/c\right)^{3d} \log(1/\delta)}{q^2 \rho^2}\right)$$

*samples from $\mathbb{P}$.*

