# OpenReview forum: "Replicable Clustering"
_NeurIPS.cc/2023/Conference — NeurIPS 2023 poster_

### Official Review · Reviewer_t2BD · 2023-07-05

**Soundness:** 4 excellent
**Presentation:** 3 good
**Contribution:** 4 excellent
**Rating:** 7
**Confidence:** 2

**Summary:**

This paper studies the design of clustering approximation algorithms in the context of statistical clustering under the notion of replicability. In replicable clustering problem, it requires that with high probability, the output of the algorithms should have the exact same partition of the sample space after two executions on different inputs drawn from the same distribution. Given some black-box $\beta$-approximation algorithm to the $k$-clustering problem, this paper gives an approximation framework that with constant probability, the output of the algorithm can achieves an $O(\beta)$-approximation satisfying $\rho$-replicable property.

**Strengths:**

This paper is the first to initiate the study of replicable clustering, which is a recent hot topic in the field of machine learning. In this paper, a new framework is introduced for handling replicable clustering requirements. This paper also establishes sampling complexities for clustering problems in different norms with different metrics. In order to achieve replicability, a novel tree decomposition method called Replicable Quad Tree is designed such that replicable estimation can be obtained based on the tree decomposition constructed. The proposed Replicable Quad Tree is interesting and is quite different from the well-known HST-based tree decomposition method, which could have great potential for designing approximation algorithms for other replicable problems.

**Weaknesses:**

1. The sample complexity has exponential dependence on the dimension $d$, which could be the main weakness for this paper when $d$ is large. Although in Euclidean metrics, this can be avoided by dimensionality reduction techniques, for general norms, there is no data-oblivious dimensionality reduction scheme.

2. The main proofs of this paper are hard to follow. It is better to provide intuitive ideas in each section before the proofs such that one could easily follow the main ideas of the proofs.

**Questions:**

1. Can the authors offer some intuitive insights into how the proposed Replicable Quad Tree ensures replicable estimation?

**Limitations:**

Since this is a theoretical paper, I don't think this paper has any potential negative societal impact.

---

> ### Author Rebuttal · Authors · 2023-08-08
>
> We thank the reviewer for recognizing the extensibility of our work and their time and effort in reading our paper. We address their comments as follows.
>
> > The sample complexity has exponential dependence on the dimension
> , which could be the main weakness for this paper when
>  is large. Although in Euclidean metrics, this can be avoided by dimensionality reduction techniques, for general norms, there is no data-oblivious dimensionality reduction scheme.
>
> We thank the reviewer for raising an important point. Although the dimensionality reduction technique only works for the Euclidean distance, we remark that the majority of clustering applications use the Euclidean distance. Therefore, this does not limit the applicability of our algorithms.
>
> > The main proofs of this paper are hard to follow. It is better to provide intuitive ideas in each section before the proofs such that one could easily follow the main ideas of the proofs.
>
> We thank the reviewer for the thoughtful suggestion and will carry it forward in the next version of our manuscript.
>
> > Can the authors offer some intuitive insights into how the proposed Replicable Quad Tree ensures replicable estimation?
>
> We can view the quad tree construction algorithm as a series of decisions based on heavy-hitter estimations. This basic statistical operation can be made replicable. Once every decision is replicable, the algorithm also becomes replicable. We will add this discussion to the next version of our manuscript.

---

> > ### Comment · Reviewer_t2BD · 2023-08-19
> >
> > I would like to thank the authors for their response and clarification. After looking at other reviews and corresponding rebuttals, I have no further questions.

---

### Official Review · Reviewer_nt3r · 2023-07-06

**Soundness:** 3 good
**Presentation:** 3 good
**Contribution:** 3 good
**Rating:** 6
**Confidence:** 4

**Summary:**

This manuscript focuses on the concept of replicability in statistical clustering algorithms. It introduces the notion of replicability, which refers to the ability of an algorithm to produce consistent results when executed multiple times on different inputs from the same distribution. Replicability is seen as crucial for ensuring the validity and reliability of scientific findings. The manuscript proposes replicable algorithms for three common clustering problems: k-medians, k-means, and k-centers. These algorithms utilize approximation routines for their combinatorial counterparts. The replicable algorithm for statistical Euclidean k-medians achieves a replicable O(1)-approximation with polynomial complexity. Similarly, the algorithm for statistical Euclidean k-centers achieves replicable O(1)-approximation with an additional O(1)-additive error but with exponential sample complexity. The manuscript also discusses the importance of clustering algorithms in unsupervised learning and the lack of a universally agreed-upon definition for the quality of clustering solutions. It highlights the challenges and trade-offs that algorithm designers face due to factors such as random initialization, similarity measures, noise in measurements, and outliers in the dataset. These issues contribute to non-replicable results in clustering algorithms. Furthermore, the manuscript provides related work on norms and parameters used in clustering and presents experimental results on synthetic distributions to validate the proposed replicable algorithms.

**Strengths:**

The manuscripts' innovation lies in proposing replicable clustering algorithms for statistical clustering problems.  It introduces the concept of replicability and highlights its significance in ensuring the validity and reliability of scientific findings.  The document aims to address concerns about replicability in clustering algorithms, which is a crucial topic in the subfields of machine learning and data science.  It presents replicable algorithms for k-medians, k-means, and k-centers problems and outlines their theoretical results.  Overall, the manuscripts' innovation lies in its contribution to the development of replicable algorithms for statistical clustering.

**Weaknesses:**

There are many theorems that are not well understood, and then there are relatively few experimental parts.

**Questions:**

1] How does the proposed replicable algorithm compare to existing clustering algorithms in terms of performance and accuracy?
2] Are there any limitations or assumptions in the proposed replicable algorithms that could affect their applicability in real-world scenarios?
3] How generalizable are the experimental results presented on synthetic distributions in 2D?  Can the replicable algorithms be applied to other types of datasets?
4] How sensitive are the replicable algorithms to the choice of parameters such as the choice of exponent in the cost function or the measure of similarity of the data?
5] Does the study discuss any potential implications or future research directions regarding the replicability of clustering algorithms?


**Limitations:**

In the experiment, real-world data sets are not used, which is not very convincing.

---

> ### Author Rebuttal · Authors · 2023-08-08
>
> We thank the reviewer for recognizing the importance of replicability in clustering as well as their time and effort in reading our paper. We address their comments as follows.
>
> > 1] How does the proposed replicable algorithm compare to existing clustering algorithms in terms of performance and accuracy?
>
> Our algorithm relies on a reduction to (weighted) clustering algorithms in the combinatorial setting. For any existing $\beta$-approximation algorithm, our techniques yield a $\beta(1+\varepsilon)$ solution for any $\varepsilon > 0$.
>
> For the case of (non-replicable) statistical k-medians/k-means, the work of Ben-David [2007] implies randomized polynomial-time constant-ratio approximation algorithms with sample complexity $O(k / \varepsilon^2)$. However, these algorithms are non-replicable and we require greater sample complexity to achieve the replicability guarantee.
>
> > 2] Are there any limitations or assumptions in the proposed replicable algorithms that could affect their applicability in real-world scenarios?
>
> For the case of k-medians/k-means, there is a blow-up in the sample complexity for non-Euclidean metrics that scales exponentially with the dimension. However, we remark that the majority of applications use the Euclidean distance where this exponential blow-up can be avoided using our dimensionality reduction method.
>
> > 3] How generalizable are the experimental results presented on synthetic distributions in 2D? Can the replicable algorithms be applied to other types of datasets?
>
> > In the experiment, real-world data sets are not used, which is not very convincing.
>
> The experimental results can be generalized to higher dimensional datasets through our dimensionality reduction technique. We presented the experimental results in 2D for simplicity of visualization and implementation.
>
> > 4] How sensitive are the replicable algorithms to the choice of parameters such as the choice of exponent in the cost function or the measure of similarity of the data?
>
> We thank the reviewer for raising an interesting question. Although we have not explored this explicitly, we expect the algorithm to be quite sensitive to the choice of parameters. For example, with regards to the exponent $p$, there is a non-trivial blow-up in the sample complexity just going from $p=1$ (k-medians) to $p=2$ (k-means). We acknowledge this is a potential direction for future research.
>
> > 5] Does the study discuss any potential implications or future research directions regarding the replicability of clustering algorithms?
>
> We believe many future directions are possible. For example, determining some lower bounds for statistical clustering, even in the non-replicable case, would be very interesting. Also, it may be possible to improve our sample complexity bounds by considering more recent coreset algorithms. Moreover, It may be fruitful to consider connections with differential privacy and other stability measures. One could also consider the problem of hierarchical clustering. In general, we believe that we have scratched the surface of this problem and hope that it inspires future research. We will add this discussion to the next version of the manuscript.
>
> [Ben-David, 2007]: Shai Ben-David. A framework for statistical clustering with constant time approximation algorithms for k-median and k-means clustering.

---

### Official Review · Reviewer_fyTj · 2023-07-07

**Soundness:** 4 excellent
**Presentation:** 3 good
**Contribution:** 3 good
**Rating:** 6
**Confidence:** 3

**Summary:**

This paper studies the concept of replicability in clustering. This topic is important because replicability is what allows for other researchers to reproduce the results of a study to verify their correctness. This is a big problem these days, 50% of scientists saying that there is a replicability crisis. In the interest of improving the scientific verification process, designing replicable algorithms (i.e., algorithms that function the same when executed twice with the same random seed). This differs from the related notion of clustering stability in that its success is more dependent on the algorithmic design than structures in the data.

In their setting, points are sampled from a random distribution over the d-dimensional unit ball according to some given metric. It models many norms, including \ell_p norms. They use Impagliazzo et al.’s notion of rho-replicability: The probability (over every two sampled sets of points and a single random seed) that the algorithm on each set but with the same random seed yields the same output is at least rho.  Outputs to these clustering problems are functions that, given a point, assigns it to a cluster (with no explicit center, just an identifier).

The main results are new algorithms for statistical k-means and k-medians, with extended improvements when using the Euclidean metric. Their approximation factors both near the approximation factor of a given black-box vanilla algorithm, and their sample complexity on general metrics has an exponential dependence on the dimension (it is sub-exponential in the Euclidean version). They also show an approximation algorithm for the replicable statistical k-centers algorithm. Given an approximation of the form a*OPT+b for k-centers, they give an approximation of the form c*OPT+d, where c and d are linear in both a and b. The query complexity is exponential in the dimensionality.

Their algorithm uses a quadtree decomposition, which creates a tree structure over all points. The leaves (which correspond to points) are then mapped to cluster centers. So when you receive a query for a point, you traverse the tree to find the leaf which contains it (as it is a partition of the space), and you map it to the corresponding cluster. However, in order to adequately depict the distribution of the space with their decomposition (which they are not given), they require access to an exponential number of queries in d. In the Euclidean case, they can first reduce the dimensionality via Johnson-Lindenstrauss, and map an epsilon-net from the original space to the low-dimensional space. There still is the difficulty of mapping the clustering between these two spaces, but it seems most of those details are in the appendix.

For replicability, they form a grid, sample points in the space, and map them to the centers of cells to estimate probability mass of each cell. On high-mass cells according to a random threshold, and then they apply a vanilla algorithm.

Finally, they compare k-means++ and k-means++ run on the core produced for their replicability results. It is pretty clear that their approach leads to much greater replicability.


**Strengths:**

This paper is written extremely clearly. I really appreciate how slow and careful the introduction and preliminaries are, which I believe could be followed by someone unfamiliar with the field. For instance, while many of us take for granted that a replicable clustering algorithm optimizes for some utility notion, they take the time to explain this and how it is distinct from their replicability goal (which is a bigger focus of their preliminary section). They also take the time to point out to the reader that their metric function and the ell_p parameter p are different, and give an in-depth but understandable explanation of the intuition behind ell_p for different values of p. The informal presentation at the beginning of results in section 3 is also very nice.

The results are nice and interesting, but I wouldn’t say they are groundbreaking. The techniques have clear foundations in previous methods, but it also seems unique how they combined them together. I wasn’t sure by reading the paper how their replicability results compare to previous literature.


**Weaknesses:**

I found it very odd that the main focus of the paper introduction and title was replicable k-centers, but they spend most of their time on (nonreplicable) k-centers. It seems like if the highlight of the paper was replicability, they should have focused on the methods used to achieve this. I feel the details of this algorithm were insufficient for a reader invested in replicable k-centers. Additionally, I wasn’t sure how their results compared to any baseline results that exist, so it is difficult to judge the novelty.


**Questions:**

1. I was a bit confused by the sentence “We emphasize that even though we only observe a sample from P we aim to solve the problem on the whole support of the distribution.” Can you clarify exactly what this means? Is it that the algorithm works on an actual given dataset P but needs to be able to work on all possible datasets? I would think that would be assumed.
2. How do your algorithms compare to existing algorithms? Is there any baseline implied by any other work?


**Limitations:**

There was no limitations section, though it did not seem needed (but would be preferred).

---

> ### Author Rebuttal · Authors · 2023-08-08
>
> We thank the reviewer for their time and effort in reading our paper and address their comments as follows.
>
> > I wasn’t sure by reading the paper how their replicability results compare to previous literature.
> > How do your algorithms compare to existing algorithms? Is there any baseline implied by any other work?
> > I wasn’t sure how their results compared to any baseline results that exist, so it is difficult to judge the novelty.
>
> To the best of our knowledge, we are the first to consider replicability as a theoretical property within the clustering setting. Thus it is not clear if there is any baseline against which we can compare our work.
>
> >I found it very odd that the main focus of the paper introduction and title was replicable k-centers, but they spend most of their time on (nonreplicable) k-centers. It seems like if the highlight of the paper was replicability, they should have focused on the methods used to achieve this. I feel the details of this algorithm were insufficient for a reader invested in replicable k-centers.
>
> Although k-centers is a well-studied clustering problem, it is not the only clustering formulation. We also study statistical k-medians/k-means clustering and provide replicable algorithms (Theorem 3.1, Theorem 3.2). Moreover, we derive replicable k-centers algorithms as well (Theorem 3.3). Our title refers to the general clustering problem and not just k-centers. We would appreciate it if the reviewer pointed out the specific parts of the introduction that lead to confusion since we do not explicitly refer to k-centers in the introduction.
>
> > I was a bit confused by the sentence “We emphasize that even though we only observe a sample from P we aim to solve the problem on the whole support of the distribution.” Can you clarify exactly what this means? Is it that the algorithm works on an actual given dataset P but needs to be able to work on all possible datasets? I would think that would be assumed.
>
> We thank the reviewer for identifying a point of potential confusion. We will clarify this in the next version of our manuscript.
>
> Clustering has mainly been studied from the combinatorial point of view, where the distribution is the uniform distribution over some finite points and we are provided the entire distribution. The statistical clustering setting generalizes to arbitrary distributions with only sample access. We wanted to clarify that although we only have access to samples, our output solution should be a good solution for the entire distribution, not just the observed data.
>
> > There was no limitations section, though it did not seem needed (but would be preferred).
>
> We will add a limitations section to summarize the assumptions that we rely on. We remark that these assumptions are explicitly stated in other sections of the manuscript.

---

### Official Review · Reviewer_Jm7K · 2023-07-18

**Soundness:** 3 good
**Presentation:** 3 good
**Contribution:** 3 good
**Rating:** 6
**Confidence:** 2

**Summary:**

The topic of this paper is replicable clustering which in high-level asks for design of an algorithm that given two runs of the algorithm on different samples from the “same” input distribution. Replicability is a notion introduced recently in a work by Impagliazzo et al. [2022].
The algorithm for statistical clustering with centroid-based objective with $p$-norm cost. For the case of $k$-means and $k$-medians, via corresponding centroid-based $k$-clustering on “finite” size (referred to as combinatorial clustering) in a black-box manner and design $O(1)$-approximation with poly($d$) sample complexity (but exponential in k?). However, the case of $k$-center is more complicated as it depends on the $\ell_\infty$-norm and very sensitive to the largest distance. The results on $k$-center has worse approximation guarantee and sample complexity and requires stronger assumptions. In particular, for the case of $k$-center, the guarantee is bicriterion.


**Strengths:**

At high-level, the notion is interesting and related to several other important topics too. The technical contribution of the paper seems to be satisfactory. However, I'm less familiar with the literature and the technicality of the paper might not be significant for someone working on this area or differential privacy.

**Weaknesses:**

The paper is following the notion of [Impagliazzo et al., 20]; however, I still not quite convinced why the shared randomness assumption is meaningful. A related question is whether an assumption such as Assumption F. 2 can be sufficient for achieving replicability?

The result seems to heavily depend on the coreset construction of Frahling and Sohler [2005]. Can you elaborate why more recent coreset construction can be applied here?

Theorem 3.1 and Theorem 3.2 provide upper bounds but there is no lowerbound. E.g., Any hope to show exponential dependence on k (or d) is needed (d, for the general metric space)?

How the algorithm in this paper is different from those for differentially private clustering?

Can you elaborate on the proof of Corollary E. 9? I particular, how does the dimension show up?


**Questions:**

Please see the questions in the weakness section.

**Limitations:**

There is no discussion of limitation by the authors. Though, the paper does not seem to have potential negative societal impact.

---

> ### Author Rebuttal · Authors · 2023-08-08
>
> We thank the reviewer for their time and effort in reading our paper and address their comments as follows.
>
> > The paper is following the notion of [Impagliazzo et al., 20]; however, I still not quite convinced why the shared randomness assumption is meaningful. A related question is whether an assumption such as Assumption F. 2 can be sufficient for achieving replicability?
>
> We thank the reviewer for raising an important point.
>
> Sharing the randomness can be thought of as a way to couple the two executions of the algorithm. Kalavasis et al. [2023] consider a notion of “replicability” where there is no need to share internal randomness and the two executions of the algorithm can be coupled in an arbitrary way. They also show that such algorithms can be converted to replicable algorithms by a specific implementation of the internal randomness. However, the conversion still requires sharing of the internal randomness and incurs an exponential computation time in the dimension of the data.
>
> For our specific setting, shared randomness is crucial in order to achieve the exact same output in two executions of the algorithm (with high probability). For numerical queries such as mean estimation, this property may not mean much, as the utility of the output (distance to true mean) naturally aligns with replicability (distance between two executions). However, for clustering, the utility (objective function) does not correlate with the replicability (difference between outputted centers along two runs). In this case, sharing randomness seems to be required for replicability.
>
> We show that Assumption F.2 is indeed sufficient for the specific case of k-centers clustering (Theorem F.8).
>
> > The result seems to heavily depend on the coreset construction of Frahling and Sohler [2005]. Can you elaborate why more recent coreset construction can be applied here?
>
> The coreset construction of Frahling and Sohler [2005] can be adapted to the statistical setting since the algorithm can be viewed as a series of decisions based on heavy hitters estimation. This elementary statistical operation generalizes from the combinatorial setting to the statistical setting. We conjecture that more recent coreset constructions can also be adapted this way if they can be viewed as a series of elementary statistical operations.
>
> > Theorem 3.1 and Theorem 3.2 provide upper bounds but there is no lowerbound. E.g., Any hope to show exponential dependence on k (or d) is needed (d, for the general metric space)?
>
> To the best of our knowledge, there is little known about lower bounds for statistical clustering, even in the non-replicable setting. We believe this would be an interesting line of future work and will mention this in the next version of our manuscript.
>
> > How the algorithm in this paper is different from those for differentially private clustering?
>
> We thank the reviewer for raising an important point.
>
> Differential privacy (DP) provides a worst-case combinatorial guarantee for two runs of the algorithm on neighboring datasets, i.e., two executions when the datasets differ in one element. Moreover, the definition of DP uses the max-divergence as a measure of statistical distance of the two posteriors of the algorithm. On the other hand, the definition of replicability we employ is statistical in the sense that it considers two executions where the inputs are (different) i.i.d. datasets from the same underlying distribution. Notice that the definition of DP does not require that the data come from a distribution, since it considers only one change in the dataset. Another difference is that instead of asking for small max-divergence between the posteriors of the algorithm on the two executions like DP, the definition of replicability requires that the outputs are exactly the same when the internal randomness is shared. Hence, these two definitions are not trivially comparable.
>
> Nevertheless, the works of Bun et al. [2023] and Kalavasis et al. [2023] provide connections between DP and replicability for certain classes of problems. Although it is not immediate from these papers, this supports the plausible conjecture that there may be a way to connect the two lines of work in the clustering setting.
>
> > Can you elaborate on the proof of Corollary E. 9? In particular, how does the dimension show up?
>
> We thank the reviewer for identifying a possible point of confusion. In general, we plan to make the appendix more self-contained in the next version of our manuscript.
>
> The main basis of our dimensionality reduction scheme is a dimensionality reduction result for the combinatorial setting (Theorem E.1). The dimension shows up as part of the statement of this result. Section E.3 extends this result to the bounded distributional setting as follows: We first discretize the bounded domain to reduce it to a weighted version of the combinatorial setting with Proposition E.8 quantifying the estimation error of such a discretization. Then Corollary E.9 follows by applying Theorem E.1 to the weighted combinatorial case.
>
> All in all, the dimension shows up since Corollary E.9 is derived from Theorem E.1 which contains the dimension.
>
> [Kalavasis et al., 2023]: Alkis Kalavasis, Amin Karbasi, Shay Moran, Grigoris Velegkas: Statistical indistinguishability of learning algorithms.
>
> [Frahling and Sohler, 2005]: Gereon Frahling and Christian Sohler. Coresets in dynamic geometric data streams.
>
> [Bun et al., 2023]: Mark Bun, Marco Gaboardi, Max Hopkins, Russell Impagliazzo, Rex Lei, Toniann Pitassi, Jessica Sorrell, and Satchit Sivakumar: Stability is stable: Connections between replicability, privacy, and adaptive generalization.

---

### Official Review · Reviewer_RdwG · 2023-07-31

**Soundness:** 4 excellent
**Presentation:** 2 fair
**Contribution:** 3 good
**Rating:** 7
**Confidence:** 4

**Summary:**

This paper initiates the study of formal replicability for clustering algorithms. Replicability is defined to be a property of an algorithm for a statistical clustering problem, requiring that fixing the internal randomness of the algorithm while resampling the input data will yield exactly the same (representation of a) mapping from points to clusters. Upper-bounds are given for the statistical k-means, k-medians, and k-centers problems, with poly(d) sample complexity proven via dimensionality reduction in the case of euclidian distance. The theoretical results are also empirically evaluated on synthetic data.

**Strengths:**

This paper addresses an interesting problem by extending recently introduced formal notions of replicability to the task of statistical clustering. Clustering seems like a setting where replicability is particularly well-motivated, as there are many reasonable desirable properties for clustering solutions, and being able to ensure that solutions which demonstrate a particular balance of these properties are in fact a result of the algorithm generating solutions, and not just a fluke of the sample, is a natural objective.

**Weaknesses:**

The writing was very readable, but it would have been nice to have a more thorough comparison of this work to the related work. What is the cost of replicable clustering compared to non-replicable algorithms for the same problems? Does this notion of replicability solve any of the issues that motivated the "stability of centers" condition for selection of k that is mentioned in Section 1.1?

I also found Section 4 a bit difficult to parse. I think some of the notation used in Algorithm 4.1 isn't defined, and though it's reasonably intuitive, using more precise language in the high-level exposition and fully defining all variables/notation would definitely improve the readability of this section.

**Questions:**

See weaknesses.

**Limitations:**

Yes, the authors address all limitations.

---

> ### Author Rebuttal · Authors · 2023-08-08
>
> We thank the reviewer for recognizing the importance of replicability in clustering algorithms. We appreciate the reviewer for their time and effort in reading our paper and address their constructive comments as follows.
>
> > What is the cost of replicable clustering compared to non-replicable algorithms for the same problems?
>
> For the case of (non-replicable) statistical k-medians/k-means, Ben-David’s work [Ben-David, 2007] implies randomized polynomial-time constant-ratio approximation algorithms with sample complexity $O(k / \varepsilon^2)$.
> For the case of (non-replicable) k-centers, ours is the first statistical formulation of the problem. Hence it is not clear how to compare our guarantees.
>
> We will add these remarks to the next version of the manuscript.
>
> > Does this notion of replicability solve any of the issues that motivated the "stability of centers" condition for selection of k that is mentioned in Section 1.1?
>
> We thank the reviewer for raising an interesting point.
>
> The work [Ben-David et al., 2006] shows that the “stability of centers” is determined by the symmetry of data and does NOT reflect anything about the choice of k. Therefore, the stability of centers is perhaps not the right criterion for selecting k. Our work reaffirms the conclusions of [Ben-David et al., 2006] in that “stability” can essentially be achieved for any choice of k.
>
> We believe that exploring more fine-grained notions of replicability, like list replicability and certificate replicability that were proposed recently in [Dixon et al., 2023], and understanding how these quantities vary as a function of k can shed light on this matter.
>
> > I also found Section 4 a bit difficult to parse. I think some of the notation used in Algorithm 4.1 isn't defined, and though it's reasonably intuitive, using more precise language in the high-level exposition and fully defining all variables/notation would definitely improve the readability of this section.
>
> We thank the reviewer for this suggestion. We will fully define all variables/subroutines used in Algorithm 4.1 within the next version of the manuscript.
>
> [Ben-David, 2007]: Shai Ben-David. A framework for statistical clustering with constant time approximation algorithms for k-median and k-means clustering.
>
> [Ben-David et al., 2006]: Shai Ben-David, Ulrike Von Luxburg, and Da ́vid Pa ́l. A sober look at clustering stability.
>
> [Dixon et al., 2023]: Peter Dixon, A. Pavan, Jason Vander Woude, and N. V. Vinodchandran. List and certificate complexities in replicable learning.

---

> > ### Comment · Reviewer_RdwG · 2023-08-18
> >
> > I thank the authors for their detailed response. I will keep my score.

---

### Decision · Program_Chairs · 2023-09-21

**Decision:**

Accept (poster)

**Comment:**

This work develops efficient clustering algorithms satisfying the recently introduced notion of replicability. Roughly speaking, replicability requires that fixing the internal randomness of the algorithm while resampling the input data will give the same mapping from points to clusters. The paper develops algorithms for $k$-means, $k$-median, and $k$-center, and also provides experimental evaluation. The reviewers agreed that this work is appropriate for publication at NeurIPS.